# Transcriptomic and open chromatin atlas of high-resolution anatomical regions in the rhesus macaque brain

Senlin Yin[1,7], Keying Lu[1,7], Tao Tan[2,3,4,7], Jie Tang [1,7], Jingkuan Wei[3], Xu Liu[5], Xinlei Hu[1], Haisu Wan[5], Wei Huang[6], Yong Fan[4]*, Dan Xie [1]* & Yang Yu [2]*

The rhesus macaque is a prime model animal in neuroscience. A comprehensive transcriptomic and open chromatin atlas of the rhesus macaque brain is key to a deeper understanding of the brain. Here we characterize the transcriptome of 416 brain samples from 52 regions of 8 rhesus macaque brains. We identify gene modules associated with specific brain regions like the cerebral cortex, pituitary, and thalamus. In addition, we discover 9703 novel intergenic transcripts, including 1701 coding transcripts and 2845 lncRNAs. Most of the novel transcripts are only expressed in specific brain regions or cortical regions of specific individuals. We further survey the open chromatin regions in the hippocampal CA1 and several cerebral cortical regions of the rhesus macaque brain using ATAC-seq, revealing CA1- and cortex-specific open chromatin regions. Our results add to the growing body of knowledge regarding the baseline transcriptomic and open chromatin profiles in the brain of the rhesus macaque.

[1] Frontier Science Center for Disease Molecular Network, State Key Laboratory of Biotherapy, West China Hospital, Sichuan University, 610041 Chengdu, China. [2] Department of Obstetrics and Gynecology, Peking University Third Hospital, 100191 Beijing, China. [3] Yunnan Key Laboratory of Primate Biomedical Research, Institute of Primate Translational Medicine, Kunming University of Science and Technology, 650500 Kunming, Yunnan, China. [4] Key Laboratory for Major Obstetric Diseases of Guangdong Province, The Third Affiliated Hospital of Guangzhou Medical University, 510095 Guangzhou, China. [5] Experiment Medicine Center, The Affiliated Hospital of Southwest Medical University, 646000 Luzhou, Sichuan, China. [6] School of Mathematics and Statistics, Key Laboratory for Applied Statistics of the Ministry of Education, Northeast Normal University, 130024 Changchun, China. [7] These authors contributed equally: Senlin Yin, Keying Lu, Tao Tan, Jie Tang *email: yongfan011@gzhmu.edu.cn; danxie@scu.edu.cn; yuyang5012@hotmail.com

The complex genetic network in the brain underlying primate behaviors, cognition and emotion has been central in neuroscience. A comprehensive transcriptomic atlas of the brain is key to a deeper understanding of the brain function, which would facilitate neurological disease studies. The transcriptomic atlas for human brains has been established, from prenatal period to adulthood[1,2]. However, less is known about the transcriptome of the rhesus macaque brain.

The rhesus macaque (*Macaca mulatta*) is phylogenetically close to human being[3], and it is one of the most commonly used non-human primate animals in neuroscience studies[4]. The structure and function of the rhesus macaque brain are similar to human brains. The neocortex (involved in higher-order brain functions) comprises 72% of the macaque brain, very close to the human neocortex that represents 80% of the brain volume[5]. In contrast, only 28% of the rat brain is neocortex. The nucleus organization and projection pathways of the macaque hippocampus are also more similar to that of human than rodents[6]. Given the similarity between human and macaque brains, a comprehensive transcriptomic atlas of rhesus macaque brain could facilitate neuroscience studies toward a better understanding of the human brain. However, previous transcriptomic studies[7–11] on the rhesus macaque brain either had limited sampled brain regions or focused narrowly on non-coding RNA.

In addition to the transcriptome, the open chromatin state is also involved in the regulation of gene expression. Region- and cell-specific patterns of open chromatin have been reported in human and mouse brain using ATAC-seq[12,13] (Assay for Transposase-Accessible Chromatin using sequencing). Nevertheless, it remains unknown whether region- or individual-associated open chromatin pattern exists in the rhesus macaque brain. Furthermore, the association between chromatin accessibility and gene expression has not been thoroughly explored in the rhesus macaque brain.

In this study, we generate an anatomically comprehensive transcriptomic atlas of the rhesus macaque brain, as well as chromatin accessibility atlas of the rhesus macaque cerebral cortex. We perform RNA-sequencing (RNA-seq) on 416 samples from 52 regions (anatomically and functionally related to human counterparts) of the rhesus macaque brain, revealing transcriptomic differences among different brain areas. We compare the transcriptomic profiles of the rhesus macaque brain with the human brain across several brain areas. In addition, we survey the open chromatin regions in several cerebral cortical regions and hippocampal CA1 (cornu Ammonis 1) of the rhesus macaque brain using ATAC-seq, revealing region-specific patterns of chromatin accessibility.

## Results

**Transcriptomes of 52 regions in the macaque brain**. To build a comprehensive transcriptome map of the rhesus macaque brain, we profiled the transcriptomes of 52 brain regions in eight rhesus macaques (416 brain samples in total, Fig. 1a–d, Supplementary Tables 1 and 2, RNA quality control information in Supplementary Note 1) using Ribo-zero RNA-seq (see Methods). Samples from the midbrain were excluded from downstream analysis except for de novo transcriptome assembly, due to possible mixture of residual tissue from the ventral tegmental area (VTA) and substantia nigra (SN) (see Methods). For each sample, we sequenced an average of 60 million raw reads (ranged 45–180 million reads, sequencing quality statistics in Supplementary Data 1), with mapping rates around 80% (ranged from 67.85 to 88.93%, Supplementary Fig. 1a–d). A total of 23,651 genes were detected (read count > 1 in at least three samples) (the RNA-seq data processing pipeline as shown in Supplementary Fig. 1e).

To gain a high-level understanding of the rhesus brain transcriptome, we visualized the samples in two dimensions using t-distributed stochastic neighbor embedding (t-SNE). The samples from non-cortical regions, like the pons (PON) in the brainstem, clustered by regions. Whereas samples from cerebral cortical regions clustered by individuals (Fig. 1e, Supplementary Fig. 1f–g and Supplementary Note 2). Such clustering patterns suggests that non-cortical structures might have less inter-individual heterogeneity than the cerebral cortex. To determine whether cellular composition contributed to inter-individual heterogeneity, we further performed deconvolution analysis based on previously published single-cell RNA-seq dataset of rhesus macaque brain[11] using CIBERSORT[14]. With the published signature of cell types from the prefrontal cortex of adult macaque, we found that different brain regions varied in inferred cellular composition (Supplementary Fig. 2a). In contrast, within each brain region, the inferred cellular composition of samples from different individuals was similar, especially in the cerebral cortex (Supplementary Fig. 2b). For instance, the cortical regions generally had abundant ExN5 (Excitatory Neuron subtype 5, Supplementary Fig. 2a, c). Meanwhile, the proportion of oligodendrocytes was especially high in the Corpus Callosum (abundant in axonal content) compared with cortical regions, which was in accordance with their difference in axonal density[15].

We also compared young (not older than 5 years) versus mid-aged (older than 5 years), and male versus female rhesus macaques using mixed-effect generalized linear model (see Methods), and identified potentially age-related (Supplementary Data 2) and sex-related genes (Supplementary Table 3) in various regions of the rhesus macaque brain (permutation-based FDR < 0.1, Supplementary Fig. 3a, b). For instance (Supplementary Fig. 3c), in the young rhesus macaques, we found upregulation of *EBF3* in the hypothalamus, which involves the development of dopaminergic neurons[16]. The brain regions of the aged rhesus macaque generally had upregulation of nicotinic cholinergic receptor *CHRNA1*, which might be a compensatory reaction to decreased acetylcholine level related to aging[17]. However, most of the age- and sex-related differential genes had not been reported so far, and further investigations are needed.

**Similar profiles between human and macaque brain**. To elucidate the conservation of regional expression profile between human and rhesus macaque brains, we compared our RNA-seq data with the human brain dataset obtained from the PsychEncode[11] (see Methods, Supplementary Tables 4 and 5). Similar to rhesus macaque, t-SNE analysis revealed that the cerebral cortex from the human brain tended to cluster by individual, while deeper structures like striatum, thalamus, cerebellar cortex clustered by region (Fig. 2a). Human and rhesus macaque brain exhibited similar expression patterns in the striatum, thalamus and cerebellar cortex. Correlation analysis of homologous genes identified the highest similarity in the same brain regions between human and rhesus macaque (Fig. 2b), especially in non-cortical regions.

Unlike the rhesus macaque counterparts, the human amygdala and hippocampus samples aggregated with the cerebral cortex samples from the same donor. The distance between the amygdala and the cerebral cortex was significantly smaller in human than in rhesus macaque. The distance between hippocampus and cortex was also smaller in human (Wilcoxon rank-sum test, both *p*-value < 1e–13, Fig. 2c). In comparison, the difference between the amygdala-cerebellum/hippocampus-cerebellum was insignificant. This suggests possible evolutionary shifting of amygdala and hippocampus to resemble function and cell composition of the cerebral cortex. We identified differential

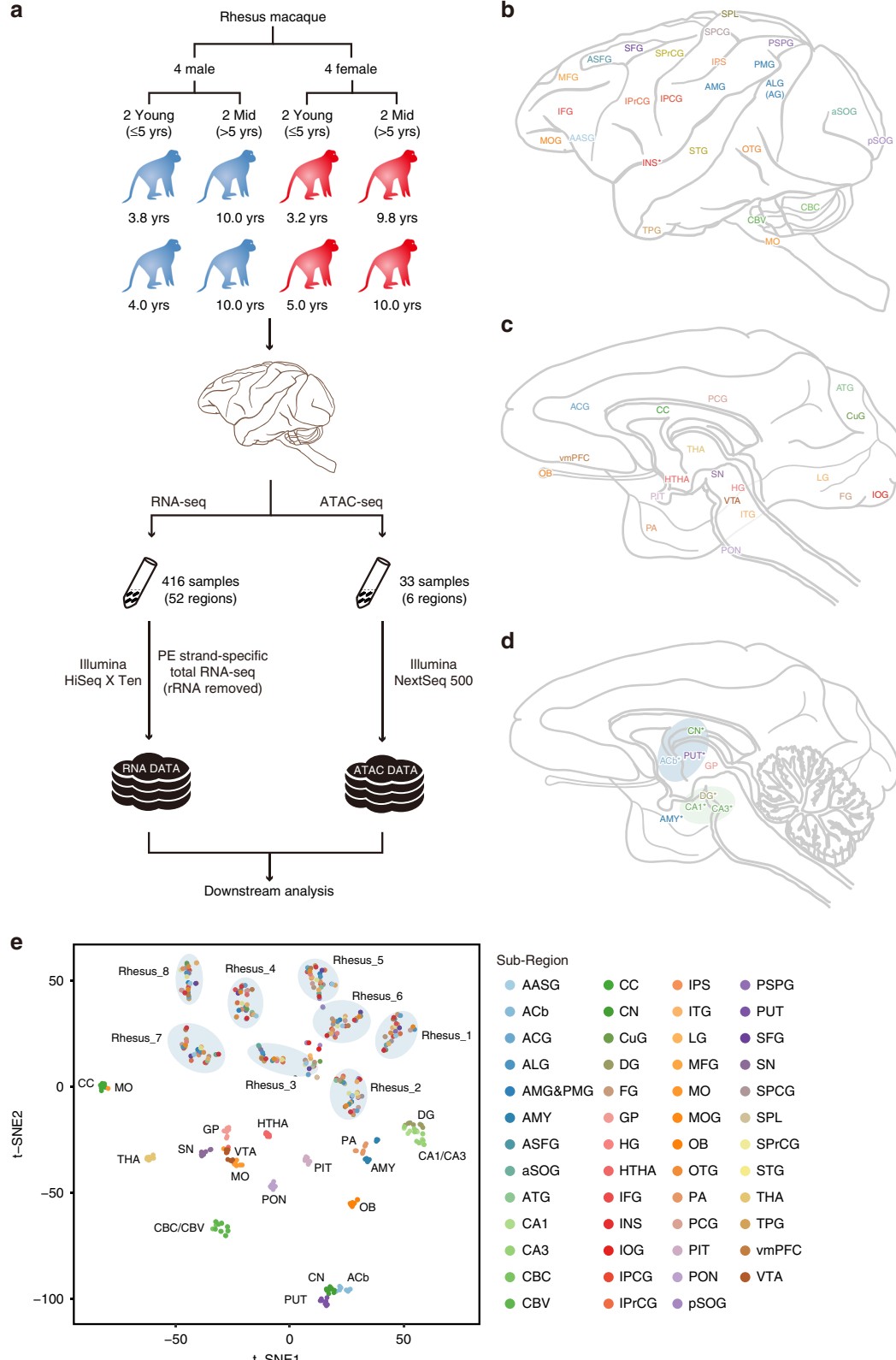

**Fig. 1 Experimental protocol, diagram of the macaque brain. a** Diagram of experimental design. **b** Lateral surface of the rhesus macaque brain, marked with name of brain regions. Full names for the abbreviations of the brain regions were listed in Table S1. **c**, **d** Median sagittal plane view of the rhesus macaque brain. **e** t-SNE scatterplot of various brain regions based on the whole-genome transcriptome, and the dot color represented its origin of region. The shadow circles showed clusters of cortex regions from each rhesus macaque individual.

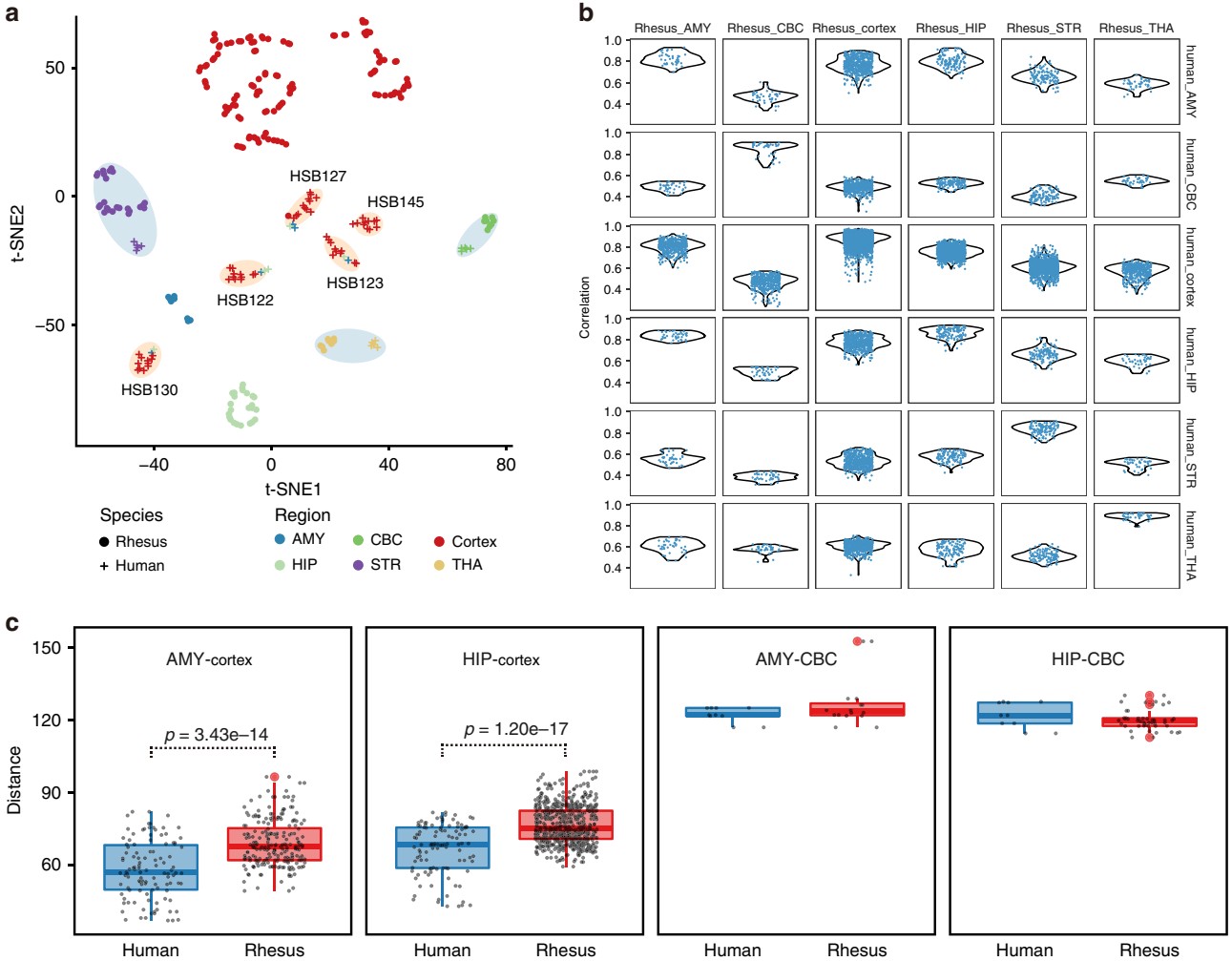

**Fig. 2 Similar expression profiles between human and macaque brains. a** Combined t-SNE using the human and rhesus macaque dataset. The blue shadow circles indicated similar expression pattern in the striatum, thalamus and cerebellar cortex between human and rhesus macaque. The orange shadow circles indicated clusters of samples from the same individual. **b** Violin plot showing the distribution of correlation coefficient of homologous genes in same structure between human and rhesus macaque. **c** Boxplot showing the euclidean distance between amygdala (AMY) and cortex, hippocampus (HIP) and cortex, AMY and cerebellar cortex (CBC), HIP and CBC in the same species. In boxplots, center line, box edges and whiskers indicate the median, upper and lower quartiles (the 25th and 75th percentiles) and 1.5 × interquartile range, respectively.

genes for each corresponding brain regions between human and rhesus macaque (Wilcoxon rank-sum test, FDR < 0.05). The highest number of differential genes was found in the cerebellar cortex (Supplementary Data 3), followed by deep structures like the amygdala and hippocampus, while the neocortex had the least inter-species differential genes, which was similar to the previous study between human and chimpanzee brains[18].

**Co-expression analysis found region-related modules**. To explore the co-expression and gene interaction network across brain regions, we identified 59 gene co-expression modules using WGCNA[19] analysis (see Methods) (Supplementary Fig. 4a). Some of the modules were highly correlated to specific brain regions (Fig. 3a), such as module M2, which was related to the pituitary (PIT) region (Pearson's r 0.987, p-value < 0.001).

The functional enrichment (adjusted p-value < 0.05, see Methods; Fig. 3b and Supplementary Table 6) and hub genes (Fig. 3c) of these modules also reflected the function of the corresponding regions. Some known marker genes were among the top hub genes of each region-related module. For instance, we found olfactory bulb-specific transcription factor *SP8*, *SP9*, and *TBX21* in the olfactory bulb-related module M11 (Supplementary

Data 4). *SP8* and *SP9* were involved in the development of somatostatin interneurons in the olfactory bulb[20,21], and expression of *TBX21* was specific to the mitral and tufted cells of the olfactory bulb in mouse[22]. We also found cerebellum-specific gene *CBLN1* and *CBLN3* (precursor of cerebellin) in the cerebellum-related module (M7).

In terms of the thalamus-pituitary axis, *AVP* (arginine vasopressin) and *OXT* (oxytocin) were among the top genes in the hypothalamus-related module (M45). Both genes are expressed in the supraoptic nucleus and paraventricular nucleus of the hypothalamus[23]. Meanwhile, the top expressed genes in the pituitary-related module (M2) were highly specific to the pituitary, including growth hormone (GH1), growth hormone-releasing hormone receptor (*GHRHR*), and anterior pituitary glycoprotein hormones common subunit alpha *CGA*.

The module M9 and M10 were composed of genes upregulated in the cerebral cortex (Supplementary Fig. 4b, c, Supplementary Data 4). For example, expression of *SATB2* (in Module M9, Supplementary Fig. 4d) is found in cortical layers II–V, and it functions as a regulator of corticocortical connections[24]. p35 protein (encoded by *CDK5R1*, in Module M10, Supplementary Fig. 4e) is a neuron-specific activator of cyclin-dependent kinase 5

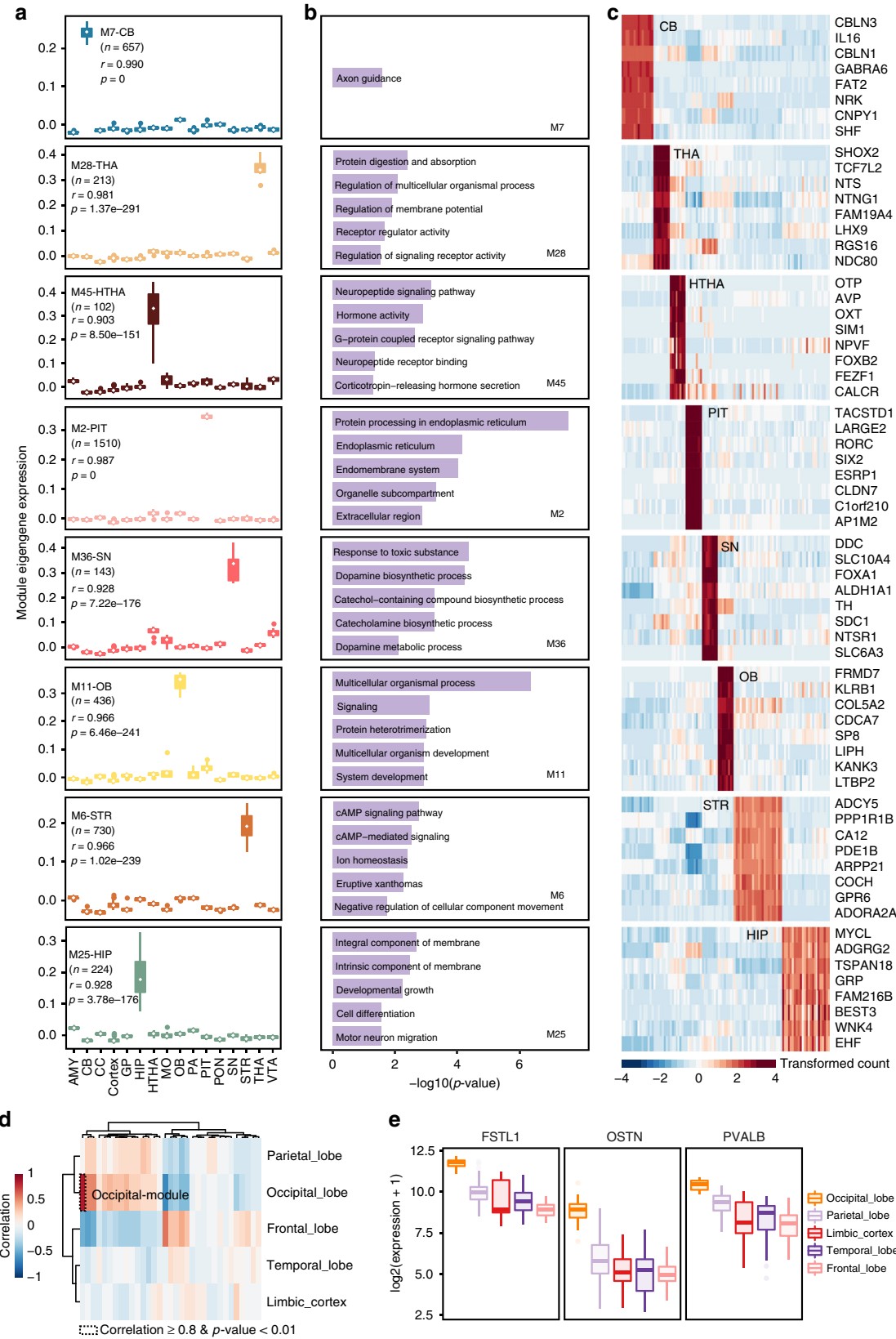

**Fig. 3 Region-related gene modules. a** Boxplot showing expression of the module eigengenes for each of the region-related gene modules. The modules were highly correlated to specific regions (Pearson's *r* > 0.9, *p*-value < 0.01). The parenthesized number represented number of genes in each module. **b** Enriched GO-terms of the corresponding gene modules. **c** Expression heatmap of the top genes in each of the region-related gene module, scaled by row. **d** Heatmap showing correlation between brain lobes and gene modules discovered in cerebral cortex samples. **e** Boxplot depicting expression of top marker genes in occipital lobe-associated gene module (occipital module). In boxplots, center line, box edges and whiskers indicate the median, upper and lower quartiles (the 25th and 75th percentiles) and 1.5 × interquartile range, respectively. The colors represented lobe of origin.

(*CDK5*), and it is essential for *CDK5* function in cortical lamination[25]. In addition, the *CDK5*/p35-regulated kinase *LMTK2* (in M9) also showed high expression in the cerebral cortex.

To examine subtle differences between cortical lobes, we next performed WGCNA on cortex samples only. We identified a gene module (Occipital module, 230 genes, Fig. 3d, Supplementary Data 5) highly correlated with the occipital lobe where the primary visual cortex lies. Several genes related to visual sensory pathway were found among the top hub genes in this module (Fig. 3e). For example, parvalbumin (*PVALB*), which was previously reported to be expressed in GABA-ergic interneurons of visual cortex[26], was ranked top. Two other top hub genes, Follistatin-like 1 (*FSTL1/OCC1*) and Osteocrin (*OSTN*), have also been reported to be associated with visual sensory activity in primate visual cortex[27].

Within the striatum, we identified genes specific to ACb/CN/putamen using the likelihood-ratio test (LRT) (Supplementary Fig. 4f). The ACb was enriched with genes involved in cell communication and signaling, suggesting a higher density of synapses. Meanwhile, the putamen was enriched with axon ensheathment, reflecting the abundance of myelinated axons. This was in accordance with previous MRI-based findings in human beings that putamen has more myelin than the caudate nucleus[28]. Such difference might be due to the difference in glial subtypes since these regions have similar glial density[29]. Regional overexpressed genes were also found in CA1/CA3/DG (dentate gyrus) of the hippocampus (Supplementary Fig. 4g) and vermis/cortex of the cerebellum (Supplementary Fig. 4h), but functional categories were not significantly enriched in these genes (gene list in Supplementary Data 6–7, and Supplementary Table 7).

**Novel transcripts related to regions and individuals**. About 30% of the RNA-seq reads could be mapped to exonic regions of the current rhesus macaque genome annotation (Fig. 4a). This exonic-mapping rate was comparable to RNA-seq dataset on human and chimpanzee brain[30], and it might be due to the insufficiency of the current annotation. Therefore, we next assembled the unannotated transcripts for each brain region using TACO[31] (Transcriptome Assemblies Combined into One, v0.7.3, assembly control and conservation analysis in Supplementary Notes 3–4). In total, we assembled 53,661 transcripts. Only 6714 (12.51%) transcripts had a perfect match in the current rhesus macaque genome annotation (Mmul 8.0.1.91). We predicted the coding potential of known and novel transcripts using CPAT[32] (Coding Potential Assessment Tool) (Supplementary Fig. 5a, b) and CPC2[33] (Coding Potential Calculator). For known transcripts, the prediction accuracy was over 95% when compared with the reference annotation (Fig. 4b). And the prediction accuracy was higher for coding transcripts than non-coding ones. As for novel transcripts, we found a good consensus between CPAT and CPC2 predictions (Fig. 4c).

Most of the assembled transcripts overlapped with exons of known genes (70.2%, 37,675/53,661, Fig. 4d). Still, a considerable proportion of the assembled transcripts were intergenic (18.1%, 9703/53,661), or intragenic (overlapped with introns of known genes, 11.7%, 6283/53,661). Among the intergenic novel transcripts, only 17.5% (1701/9703) intergenic transcripts were predicted as coding transcripts. With our annotation, over 52% of the total reads fell within exons while only <18% were intergenic (Supplementary Fig. 5c).

To explore their potential function, these novel coding transcripts were aligned to human protein sequences from Swiss-Prot (Supplementary Data 8) after in silico translation (see Methods). A total of 222 transcripts were highly conserved with human protein sequences. Eight of these transcripts (*ZNF816*, *ENO1*, *IGBP1*, *VAPB*, *MAD2L1*, *MRPL40*, *ARF1*, *TOMM20*) were annotated at different locations (Mmul 8.0.1.91) with mismatches (Supplementary Table 8). Some of these transcripts had varied expression across different brain regions (Supplementary Fig. 5d), suggesting the potential presence of pseudogenes with active transcription in these regions.

We next compared the expression level of novel transcripts in different regions of the rhesus brain. We found that TU12213 (on chromosome 12) had extremely low expression in the frontal lobe (Fig. 4e), compared with other cortical lobes. We also found some transcripts that were silent across the entire cerebral cortex. For example, TU31016 (the same protein sequence as *GPR139*) had a prominent expression in the striatum, medulla oblongata, hypothalamus, and VTA, but it was almost silent in the pituitary and cerebral cortex (Fig. 4f). *GPR139*, which was recently recognized as a receptor for *ACTH* and melanocyte-stimulating hormone (*MSH*)[34], had well-known signaling activity in the above regions. In addition, we identified several region-associated novel transcripts using LRT (Supplementary Fig. 5e), including *DLX2* (TU11217) and *DLX5* (TU34843) in olfactory bulb, *MRGPRE* (TU14465) in pons, GABA receptor *GABRQ* (TU53319) in amygdala and hypothalamus, along with cholecystokinin receptor *CCKAR* (TU39329) in substantia nigra. The absence of these subcortical transcripts in the current annotation was likely due to the fact that previous RNA-seq studies focused on the cerebral cortex of the rhesus macaque brain.

Our data confirmed several previously predicted genes[10]. An interesting case was TU44651, which had 99% similarity with the human gene *ISL2* (insulin gene enhancer protein), with only one mismatched amino acid residue (Fig. 4g). In human tissues, the expression of *ISL2* is highly specific to the pituitary and salivary gland[35], which coincided with our finding that TU44651 (*ISL2*-like transcript) was specifically expressed in the rhesus macaque pituitary. This explained why the previous study using the caudate nucleus and the cerebral cortex[10] failed to detect transcripts of *ISL2* in the rhesus macaque brains.

**Novel lncRNAs related to brain regions**. LncRNAs played an important role in neurodevelopment and brain function[36]. However, lncRNAs were poorly represented in the current annotation of the rhesus macaque genome. We identified 2845 novel lncRNAs (Methods, Supplementary Data 9) with low coding potential and length > 200 nt. With WGCNA, we identified seven lncRNA modules, five of which showed a high correlation with specific brain regions (Pearson's $r > 0.8$ and $p$-value < 0.01, Supplementary Fig. 6a, b).

Previous studies suggested that lncRNA might function as a decoy for transcription factors, especially RNA-binding proteins (RBPs), and therefore inhibit their activity[37]. We scanned the lncRNA sequences in each of the lncRNA modules for the enrichment of known RBP-binding motifs and discovered module-specific motifs (Supplementary Fig. 6c). For instance, the lncRNAs in the cerebellum-associated module were enriched in motifs for *RBM38* and *ESRP2*, while the cerebral-specific lncRNAs were enriched in binding sites for *PTBP1*, which is a repressor of the neural-specific splicing program and crucial for neuronal differentiation[38]. The full list of region-specific novel lncRNA modules could be found in Supplementary Data 10.

**Region-related chromatin accessibility patterns**. Previous studies suggest that stimuli could alter gene expression via changes in chromatin accessibility in neurons[13], which could affect synaptic plasticity, learning, and memory[39]. We performed

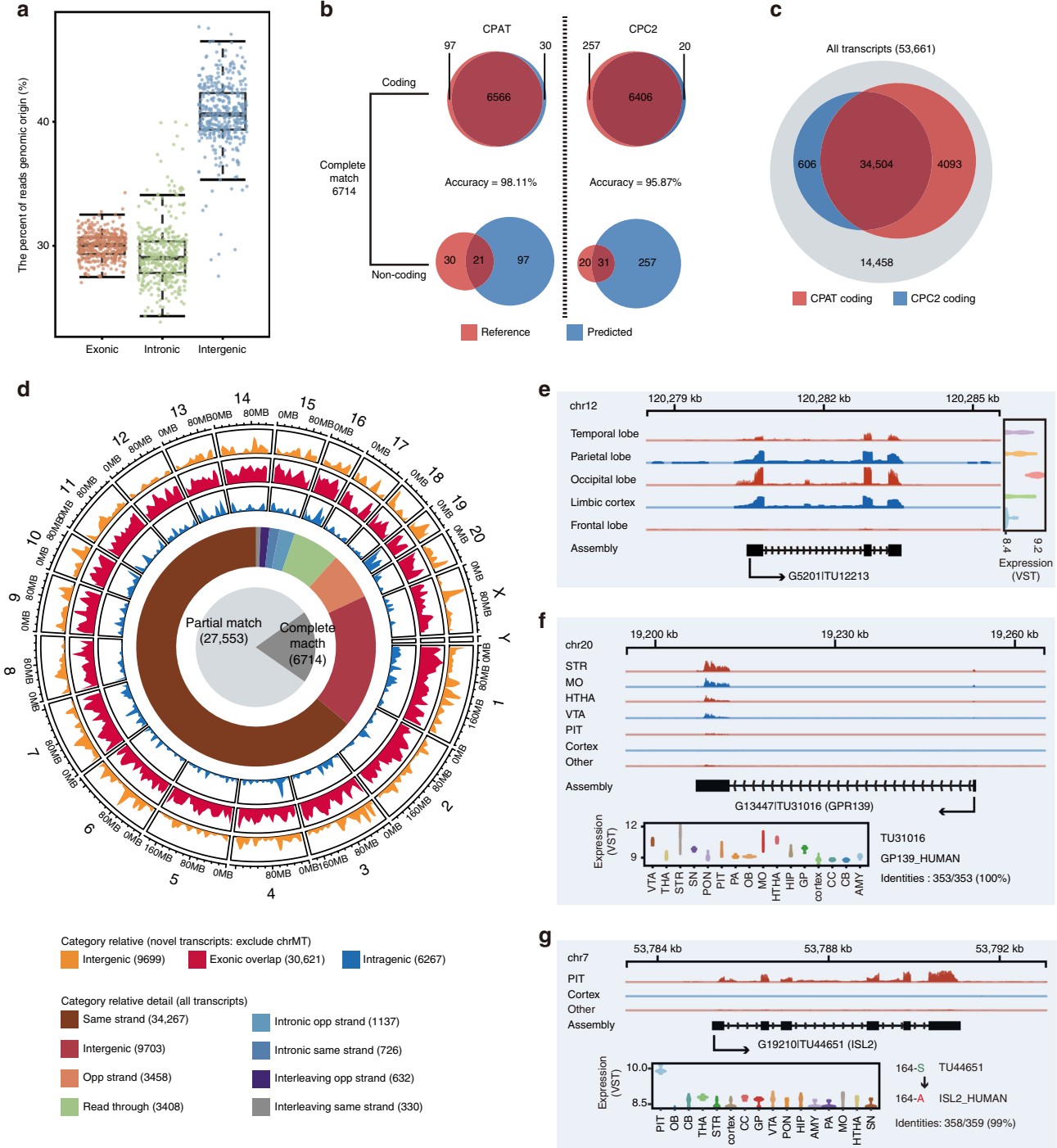

**Fig. 4 Novel transcripts related to brain regions and individuals. a** Boxplot showing the percentage of RNA-seq reads mapped to different genomic regions on the rhesus macaque genome. In boxplots, center line, box edges, whiskers, and points indicate the median, upper, and lower quartiles (the 25th and 75th percentiles), 1.5 × interquartile range and outlier, respectively. **b** Venn plot showing the accuracy of CPAT and CPC2 predicted coding potential comparing with the reference transcripts. **c** Venn plot showing the number of coding transcripts with coding potential predicted by CPAT and CPC2. **d** Circos plot shows the distribution of assembly transcripts. The pie chart shows 12.51% transcripts were complete match with reference transcripts. The donut chart shows the composition of the assembled transcripts relative to reference transcripts. The three circles of area charts indicate density of novel intergenic (orange), exonic_overlap (red), and intragenic (blue) transcripts on the genome, respectively. **e** Genome browser view showing lobe-related expression of novel transcript TU12213. **f** Genome browser view showing subregion-related expression of novel transcript TU31016 that is homologous with human protein GPR139. **g** Genome browser view showing subregion-specific expression of novel transcript TU44651, located on chromosome 7.

ATAC-seq to explore open chromatin profiles in the rhesus macaque brain (Supplementary Fig. 7a). We identified ATAC peaks for ITG (inferior temporal gyrus), MFG (middle frontal gyrus), PCG (posterior cingular gyrus), pSOG (posterior superior occipital gyrus), and SPL (superior parietal lobule) regions in the cerebral cortex, as well as the hippocampal CA1 region, in a total of 33 samples (see Methods). The ATAC-seq results were reliable considering short fragment length (Supplementary Fig. 7b), and high mapping rates (over 97.67%, Supplementary Fig. 7c, and Supplementary Table 9). Only 1.5–15.6% (mean 5.3%) of the ATAC-seq reads could be mapped to the mitochondrial genome (MT), which is comparable to previous ATAC-seq studies[40,41].

ATAC-seq peak calling was then performed for each region (see Methods). We first compared the genomic feature distribution of reproducible peaks shared between multiple brain regions. We discovered that region-specific peaks (found in only one region, noted as reproducible 1) were generally intergenic or intronic. Meanwhile, more conserved ATAC peaks (shared by multiple regions, reproducible 2–6) tended to enrich in promoters. Only 7.4% of all consensus peaks were open in all six regions. Most of these peaks (58.99%) were within 1 kb range of TSS, suggestive of constitutive promoters (Fig. 5a). We further compared the chromatin accessibility profile of hippocampal CA1 with PCG of the cerebral cortex using DiffBind and discovered 268 ATAC peaks with higher accessibility in CA1 and 730 peaks with higher accessibility in PCG (Fig. 5b and Supplementary Fig. 7d, log2FC > 1, BH-adjusted FDR < 0.1, Diffbind). Most of these differential peaks were located in introns or distal intergenic regions, suggestive of enhancers (Supplementary Fig. 7e). Therefore, we then examined the overlap between the differential peaks and known cortical and subcortical enhancers in macaque brain[42]. We found that 23.9% of hippocampal CA1-open peaks overlapped with known subcortical enhancers. In contrast, only 16.9% of the consensus peaks fell within subcortical enhancers (chi-square test $p$-value < 0.001, Fig. 5c). As for cortical enhancers, the majority of PCG-open peaks (~60%) overlapped with known cortical enhancer, whereas only 16% of the consensus peaks fell within cortical enhancers (chi-square test $p$-value < 0.001, Fig. 5d).

To determine the function of these putative enhancers, we identified 1344 genes associated with CA1-open peaks and 1908 genes associated with PCG-open peaks, based on RNA-expression data (Supplementary Data 11 and 12). The genes associated with CA1-open peaks were mostly involved in development (Fig. 5e), which could be explained by active neurogenesis from prenatal to adult hippocampus[43]. As previously reported, NEUROD1 is essential for adult neurogenesis in the hippocampus[44]. We discovered significant correlation (Pearson's $r$ 0.78, FDR 0.08, Fig. 5f) between the upregulation of NEUROD1 (log2FC 2.13) and an open ATAC peak (chr12:43455941–43456219, log2FC 1.71) in CA1. Conversely, the genes associated with PCG-open peaks were enriched in cell communication and synaptic signaling (Fig. 5e), suggesting a higher density of synapses in the cortex. For instance, we observed correlation (Pearson's $r$ 0.88, FDR 0.07, Fig. 5g) between upregulation of MCHR2 (Melanin Concentrating Hormone Receptor 2, log2FC 6.16) and open peak (chr4:155317171–155317425, log2FC 2.51) in PCG. We further scanned the differential motif enrichment between CA1-open and PCG-open peaks and revealed distinct enrichment of TF-binding motifs (Supplementary Fig. 7f). For instance, we found enrichment of motifs binding to BCL11A in CA1-open peaks. Expression of BCL11A has been found in the hippocampus and it interacts with TLX in regulation of maintenance and self-renewal of neural stem cells in the hippocampus[45]. We identified motifs that bind to cortex-related TFs like Early Growth Response gene family[46] (EGR1, EGR2, EGR3) and neuronal PAS gene family[47] (NPAS1, NPAS2). Our data suggest that the open

ATAC peaks that were common amongst different brain regions largely represented promoters, while differential peaks between CA1 and PCG were mostly intergenic or intronic, representative of distal regulatory elements, which were associated with TFs and genes related to the hippocampus and the cortex.

## Discussion

This study provided comprehensive insight into the transcriptome and chromatin accessibility of various brain regions in healthy rhesus macaque, combining RNA-seq with ATAC-seq. Previous transcriptomic studies were mostly longitudinal and focused on the development of rhesus brain[7,11], and were limited in the number of brain regions[8–10]. In contrast, we profiled the transcriptome of 52 rhesus brain regions, achieving a transcriptomic map with much higher resolution.

We identified genes with brain region-specific expression, especially in deeper brain structures. We also found that the transcriptomes of different cortical regions were more similar within each individual than between individuals, and the opposite was true for lower brain structures such as pituitary, cerebellum, striatum, thalamus, etc. where strong similarities were within each region. Similar findings have been reported in transcriptomic studies of human brains using microarray profiling[48,49]. In this study, we carefully controlled the environmental factor and conditions of the rhesus macaques, so as to minimize the potential confounding factors in human studies. We further compared the rhesus macaque RNA-seq data with previously published data on human brains. Contradicting to the previous findings, correlation of orthologous genes expression between human and rhesus macaque were high in our AMY and HIP samples. However, previous comparison studies between human and rhesus macaque/chimpanzee were mostly based on poly-A RNA-seq[11,18,50], focusing only on mature mRNA in the brain. It has been reported that the frontal cortex of the human brain had a significantly higher proportion of immature transcripts than liver[30]. Therefore, the immature transcripts in the brain might account for the difference between our study and previously reported results.

We identified a considerable amount of novel coding and non-coding transcripts that were associated with individual or brain regions. These genes had been ignored in previous annotation studies due to expression limited only in certain brain regions[10,51]. Our study further revealed a difference in chromatin accessibility profile between the cerebral cortex and the hippocampus of rhesus macaque using ATAC-seq. The differential open chromatin regions were mostly intergenic or intronic, representative of putative enhancers, which were associated with TFs and genes related to the hippocampus and the cortex. Our results resonated with the known cortical and subcortical enhancers in rhesus macaque brain[42], and provided insight into the hippocampus-related enhancers. Through the combined analysis of chromatin accessibility and gene expression, our data could contribute to a deeper understanding of genetic regulation in the hippocampus and cerebral cortex of the rhesus macaque brain.

Pooling RNA-seq data from 52 brain regions and 416 samples, along with ATAC-seq data of the hippocampus and the cerebral cortex, we generated a substantial dataset on the brains of the rhesus macaque. Overall, this study added to the growing body of knowledge regarding the normal baseline transcriptome and open chromatin profiles in the brain of rhesus macaque and provided a valuable resource for further neuroscience studies.

## Methods

**Experimental design and animal care.** Eight healthy rhesus macaques with balanced age and sex from a single source were used in this study. The eight rhesus macaques were all originated in Southern China and had similar rankings in the group. The macaques were group-housed and lived outdoors. Individuals with the

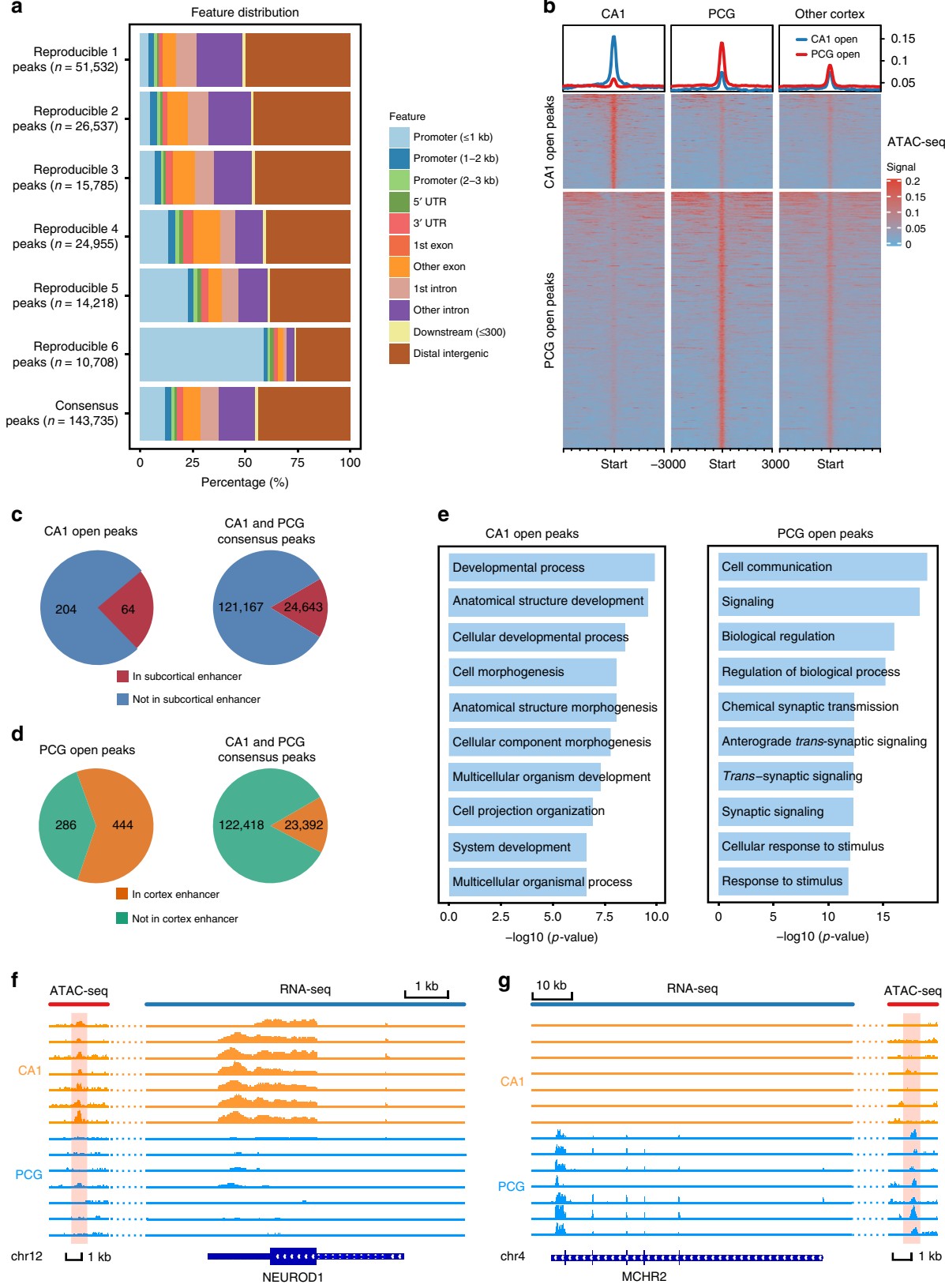

highest or lowest rankings were excluded from this study. The procedure of execution and brain harvest was standardized. For each of the rhesus macaques, the brain was harvested as soon as the rhesus macaque was euthanized to minimize the ischemic time. The brain was then cooled to 4 °C and carefully dissected. The dissected samples were stored in liquid nitrogen before RNA extraction.

During this study, we complied with all relevant ethical regulations for animal testing and research, including the Animal Ethics Procedures and Guidelines of the People's Republic of China. This study was approved by the Animal Ethics Committee of West China Hospital, Sichuan University.

**Brain dissection and sample processing.** For each individual, the brain was dissected into 52 regions by one professional animal anatomist with thorough knowledge of the rhesus macaque brain structures. The time used to harvest each region was

**Fig. 5 Differential analysis of chromatin accessibility. a** Accumulated barplot showing the feature distribution of reproducible peaks and consensus peaks of all six regions (reproducible X means peak found in X regions). **b** Heatmaps depicting enriched ATAC-seq signal around cornu Ammonis 1 (CA1)-posterior cingular gyrus (PCG) differential peaks in CA1, PCG and other cortical regions. At the top, the enriched lines showing mean values of ATAC-seq signals around the differential peaks (CA1-open and PCG-open) in CA1, PCG and other cortical regions. **c** Pie charts showing the counts of CA1-open peaks (left) and consensus peaks (right) overlapping with known subcortical enhancers. **d** Pie charts showing the counts of PCG-open peaks (left) and consensus peaks (right) overlapping with known cortical enhancers. **e** Barplot showing the function enrichment of the genes correlated with the CA1-open peaks (left) and PCG-open peaks (right) (Pearson's r > 0.7, FDR < 0.1). **f** Genome browser view showing ATAC-seq signal and RNA-seq expression profiles around the *NEUROD1* loci. The orange shadow represented the CA1-open peak that was correlated with the *NEUROD1* expression. **g** Genome browser view showing ATAC-seq signal and RNA-seq expression profiles around the *MCHR2* loci. The orange shadow represented the PCG-open peak that was correlated with the *MCHR2* expression.

---

between 30 and 60 s, and dissection of 52 brain regions from one brain was completed in about 30 min. For each region, about 2–3 mm$^3$ brain tissue were removed. The anatomical landmarks in the rhesus monkey brain atlas (A Combined MRI and Histology Atlas of the Rhesus Monkey Brain in Stereotaxic Coordinates 2nd Edition) and the Brain Maps[52] (available at http://www.brainmaps.org) were used to locate samples in rhesus macaque brains. The dissection of the brain was carefully performed according to the dissection protocol (Fig. 1b–d, Supplementary Table 1).

In specific, the midbrain was defined as the remaining tissue between diencephalon and pons, after extraction of VTA and SN. To ensure sampling accuracy, only the center of VTA and SN was sampled, possibly leaving residual tissue in the midbrain samples. Therefore, the midbrain samples were excluded in the downstream differential analysis.

**RNA extraction and RNA-seq library preparation.** A total amount of 3 μg RNA per sample was used as initial material for RNA sample preparations. RQ1 DNase (Promega) was first applied to remove DNA. RNA degradation and contamination were detected by 1% agarose gels; then, RNA purity was checked using the kaiao K5500® Spectrophotometer (Kaiao, Beijing, China). The RNA integrity and concentration were assessed using the RNA Nano 6000 Assay Kit of the Bioanalyzer 2100 system (Agilent Technologies, CA, USA).

Ribosomal RNA was removed using Epicentre Ribo-Zero™ Gold Kits (Human/Mouse/Rat) (Epicentre, USA). Subsequently, the sequencing libraries were generated following manufacturer recommendations with varied index label by NEBNext® Ultra™ Directional RNA Library Prep Kit for Illumina (NEB, Ispawich, USA).

The details of library construction could be described as follow: First, RNA fragmentation was carried out by NEBNext First Strand Synthesis Reaction Buffer under elevated temperature at 94 °C for 15 min. Subsequently, the first cDNA strand was synthesized using random hexamer primers (primers listed in Supplementary Note 5) and RNA fragments as a template. Second strand cDNA synthesis was then performed using buffer, dNTPs, DNA polymerase I, and RNase H. The library fragments were purified with QiaQuick PCR kits and eluted with EB buffer. Then terminal repair and sequencing adaptor addition were implemented. In order to select cDNA fragments of preferentially 300 bp in length, the library fragments were purified with agarose gel electrophoresis and the UNG enzyme was used to digest the second strand of cDNA. PCR was performed, aimed products were retrieved by agarose gel electrophoresis, and the library was completed. After preparation, the libraries were then sequenced on the Illumina HiSeq X Ten (Hiseq × Ten Reagent Kit v2.5) platform.

**Nuclei isolation and ATAC-seq experimental protocol.** The brain tissue was minced into tiny cubes < 0.5 mm$^3$ with a scalpel and transferred into a 15 mL conical tube (BD Falcon) containing brain tissue digestion medium (8 mL pre-warmed hibernate-E medium, 1 mg/mL collagenase I and 0.5 mg/mL collagenase IV, 1 mg/ml papain, 10 U/μl DNase I). The tissue was digested on Tube Revolver (0.3 × g) for 10 min at 37 °C. The cell suspension was pipetted further to disperse it into single cells and then filtered using a 70-μm nylon mesh (BD Biosciences). The suspension was rinsed with 20 ml PBS twice and immediately placed on ice. After centrifuging at 300 × g at 4 °C for 5 min, the supernatant was discarded and the cell pellet was resuspended in pre-cooled 1X PBS (without calcium and magnesium) for counting.

Standard ATAC-seq[53] was performed. First, to prepare nuclei, we spun 10,000 cells in a 1.5-ml microcentrifuge tube and centrifuged at 500 × g for 5 min at 4 °C, which was followed by the supernatant being removed without disrupting the cell pellet. The cells were washed once with 50 μl of cold PBS buffer, centrifuged 5 min at 500 × g, 4 °C. Then we removed supernatant. Cells were lysed using 50 μl chilled lysis buffer (10 mM Tris-HCl, pH 7.4, 10 mM NaCl, 3 mM MgCl$_2$, and 0.1% IGEPAL CA-630). Immediately after gently pipetting up and down to resuspend the cell pellet, nuclei were spun at 500 × g for 10 min using a refrigerated centrifuge at 4 °C. To avoid losing cells during the nuclei preparation, we used a fixed-angle centrifuge and carefully pipetted away from the pellet after centrifugation. Immediately following the nuclei preparation, after removing the supernatant without disrupting the nuclei pellet, the pellet was resuspended in the transposase reaction mix (25 μL 2 × TD buffer, 2.5 μL transposase (Illumina) and 22.5 μL

nuclease-free water). The transposition reaction was carried out for 30 min at 37 °C. Directly following transposition, the sample was purified using a Qiagen MinElute kit. Following purification, we amplified library fragments using 1 × NEBnext PCR master mix and 1.25 μM of custom Nextera PCR primers 1 and 2[53] (primers listed in Supplementary Note 5), under the following PCR conditions: 72 °C for 5 min; 98 °C for 30 s; and thermocycling at 98 °C for 10 s, 63 °C for 30 s, and 72 °C for 1 min. The libraries were purified using a Qiagen PCR cleanup kit yielding a final library concentration of ~30 nM in 20 μL. Libraries were amplified for a total of 10–15 cycles. Libraries were sequenced on Illumina NextSeq 500 (Next500 kit v2 High Output 150 cycles).

**RNA-seq processing, quality control, and normalization.** The pipeline for RNA-seq processing was described in Supplementary Fig. 1e. The fastq files generated by the Illumina workflow underwent read-level quality control before mapping, using FastQC (v0.11.4). For samples with poor read quality, quality trimming was performed using Trim Galore (v0.4.4, Quality Phred score cutoff: 30). The HISAT2[54] (v2.0.5, default parameters) was utilized for mapping the reads with the Mmul 8.0.1 reference genome and the Mmul 8.0.1.91 transcriptomic annotation GTF (Ensembl, available at http://ftp.ensemblorg.ebi.ac.uk/pub/release-91/gtf/macaca_mulatta/Macaca_mulatta.Mmul_8.0.1.91.gtf.gz). Read counts of genes were generated using HTSeq (v0.6.1), and the read count matrix for transcripts was assembled by prepDE.py script from the StringTie (v1.3.4, with parameter -e) package.

To evaluate RNA integrity at the transcript level, TIN (transcripts integrity number)[55] was calculated for each sample using tin.py from RSeQC (v2.6.4) tools. The median TIN score (medTIN) across all transcripts was an accurate and reliable measurement of RNA integrity[55].

A total of 23,651 genes (raw read count higher than 1 in at least three samples) were detected and used in downstream analysis. Normalization of the read count matrix was achieved by DESeq2 (v1.18.1). For visualization, the count data were first transformed using variance stabilizing transformation (VST) function from DESeq2. Then dimension reduction was performed using Rtsne (v0.13, R package).

For genome browser visualization, samtools (v1.35, parameters: view -bf 0x2) was used to filter paired reads from mapped bam file and converted into bedgraph using bedtools genomecov (v2.25.0, parameters: -split -bg). Finally, UCSC bedGraphToBigWig was used to produce bigWig for each of the RNA-seq data.

**Single-cell deconvolution analysis.** Deconvolution analysis was performed using CIBERSORT[14] (available at https://cibersort.stanford.edu/), based on previously published cell-type signatures from single-nucleus RNA-seq dataset of rhesus macaque brain[11] (available at http://evolution.psychencode.org/files/processed_data/snRNA-seq/rhesus_adult_snRNA-seq_celltype_signature.xlsx). According to previously reported method[11], the top 24 genes ranked by Preferential Enrichment Measure (PEM) score were selected per cell type to create the signature mean expression matrix. The normalized count matrix in our study and the signature matrix were imported into CIBERSORT. Then, the abundance of cell types in each sample was estimated with default parameters of CIBERSORT. For each region (one sample from each individual, a total of eight samples), the pairwise Pearson's correlation of cellular composition between samples was calculated.

**Age and sex-related differential gene analysis.** A mixed-effect model-based method was applied to analyze our dataset. Mathematically, this model, called the Negative Binomial mixed-effect generative model (NBMEGM), can be written as:

$$Y_{gijkls} \sim NB(\mu_{gijkl}, \theta),$$
$$\log(\mu_{gijk}) = \alpha_{gi} + \beta_{gj} + \gamma_{gk} + \delta_{gl},$$
$$g = 1, \ldots, G; i = 0, 1; j = 0, 1; k = 1, \ldots, 16; l = 1, 2; s = 1, 2, 3.$$

where $Y_{gijkls}$ represent the $k$-th observed number of sequencing reads of gene $g$ from individual $l$ at sex level $i$, age level $j$, region $k$, $\mu_{gijkl}$ represent the average parameter of the Negative Binomial distribution with $\theta$ measuring the rate of over-dispersion, $\alpha_{gi}$, $\beta_{gj}$, and $\gamma_{gk}$ are fixed effects, and respectively, denote the effect of age, sex, and region at level $i$, $j$, and $k$ for gene $g$, $\delta_{gl}$ is a random effect for gene $g$

from individual $l$ and follows normal distribution $N(0, \sigma_{gl}^2)$. We here assume that $Y_{gijkls}$ are mutually independent from each other.

In order to avoid the cross-impact of age and sex in detecting the covariant related genes, we implemented permutation-based large-scale hypothesis test[56] to identify genes that are related to sex and age. For sex-related gene identification, we pooled the four measurements of each gene within the $j$-th age level, permuted them and assigned each two reordered measurements to the cells $(i, j)$ for $i = 0, 1$. Here cell $(i, j)$ means a dataset with sex at level $i$, age at level $j$. This procedure was repeated for $j = 0, 1$ to complete one permutation. We conducted all 18 permutations for each gene dataset, fitted NBMEGM to each permuted dataset by using gam method (R package mgcv) and obtained a total of $18 \times G$ $p$-values. These $p$-values are then used to construct the empirical null distribution. The empirical null distribution that constituted by directly pooling the resulted $p$-values can always be biased, as mentioned by these studies[56–58]. This is mainly because this distribution is a permutation distribution under the null hypothesis mixed with the one under the alternative hypothesis, thus could induce a larger variation. We, therefore, applied locfdr method (R package locfdr) proposed by Efron et al.[59], to adjust the bias brought by the non-null $p$-values and to give an estimation to the real empirical null distribution. The significant genes were then selected according to the standard FDR method based on this estimated empirical null distribution, using fdrci (R package, FDR < 0.1, log2FC > 1). Identification of age-related genes followed the same technical line.

**Gene module discovery**. The normalized count matrix was generated using the counts function from DESeq2. goodSamplesGenes (WGCNA, v1.63) was used to eliminate genes with too many missing samples or zero variance. After pre-processing, a total of 23,651 genes measured across 408 samples were left for WGCNA-based gene module discovery. A soft threshold of 6 was chosen using pickSoftThreshold command, and a minimal number of genes per module of 50 were used for one-step network construction and module detection using blockwiseModules.

The gene modules were characterized based on the expression of the module eigengene (ME), and region traits were expressed as a binary matrix. Pearson correlation between each region and each ME was calculated, and the student asymptotic $p$-value was calculated correspondingly. The correlation between a region and a gene module was considered significant if the correlation coefficient was no <0.8 and the $p$-value was < 0.01.

The intramodular connectivity (KIM) was calculated for each of the genes within the highly correlated gene modules using intramodularConnectivity function in WGCNA. Genes with high KIM were considered as representative genes (hub genes).

The samples from the cerebral cortex were used for WGCNA-based discovery of lobe-specific gene modules, with methods described above. A soft threshold of 9 was chosen using pickSoftThreshold.

**Identification of subregion-specific genes**. To find sub-regions specifically expressed genes within STR, HIP, and CB, the likelihood-ratio test (nbinomLRT in DESeq2 package, based on negative binomial distribution) was performed comparing full model ~Region + ID with reduced model ~ID. To correct for multiple hypothesis test, Benjamini and Hochberg approach was used to adjust $p$-values (pAdjustMethod BH, DESeq2). Genes with FDR < 0.05 and fold-change > 2 compared with all other regions (pairwise comparison) were used for functional annotation.

**Comparison with human brain RNA-seq datasets**. The RNA-seq counts matrix for the human brain was acquired from PsychEncode Human brain evolution (downloaded from http://evolution.psychencode.org/). Regions from the cerebral cortex (A1C, DFC, IPC, ITC, M1C, MFC, OFC, S1C, STC, V1C, and VFC), the amygdala, the cerebellar cortex, the hippocampus, the thalamus (MD), and the striatum were included. Individuals were included if no missing regions were found in the above regions. A total of 5 human individuals and 80 samples were included for comparative analysis.

RNA-seq data from the corresponding regions of rhesus macaque (184 samples) were used for comparative analysis. The orthologous gene list between human and rhesus macaque was acquired from BioMart Ensembl[60] and 19,810 one-to-one orthologous genes were used for downstream analysis. DESeq2 and limma (removeBatchEffect, v3.34.9) were used for cross-species normalization, and to minimize the batch effect. The top 1000 genes with highest mean expression were selected to calculate correlation between rhesus macaque and human.

The euclidean distance between AMY/HIP and other regions was calculated within individual. The differential expression genes between human and rhesus macaque were computed for each region using Wilcoxon rank-sum test (BH-adjusted FDR < 0.05, |log2FoldChange| > 1).

**Novel assembly of transcriptome**. The mapping results of each sample were integrated to assemble a refined transcriptome using stringtie (v1.3.4) and TACO[31] (v0.7.3). For optimal outcomes, TACO was repeated with a different threshold for --filter-min-expr parameter (from 0.5 to 9.5 at the interval of 0.5) limiting the minimal expression allowed for the assembled transcripts.

Gffcompare (v0.10.4) was used to compare each of the assembled GTF files with the reference transcriptome (Mmul 8.0.1.91), and to estimate the accuracy of the de novo transcriptome assembly by assessing the sensitivity and precision of assembly at the exon and intron levels. F1 score was used to express both the precision and sensitivity, defined as follows:

$$F1 = \frac{2 \cdot \text{Sensitivity} \cdot \text{Precision}}{\text{Sensitivity} + \text{Precision}}$$

For exons, maximal F1 score was achieved when --filter--min-expr was 6.5; Meanwhile, for introns, maximal F1 score was achieved when --filter-min-expr was 4.5. Therefore, --filter-min-expr was set to 5.5 for optimal de novo assembly.

**Analysis of coding/orthologous transcripts**. The coding potential of the novel transcripts was estimated using Coding Potential Calculator[33] (CPC2) and Coding Potential Assessment Tool[32] (CPAT v1.2.4) with default parameters. For CPAT, the optimal cutoff value for coding probability (CP) was determined with a two-graph ROC curve. To achieve maximal specificity, a transcript was considered as with coding potential only if both CPC2 and CPAT predicted it as coding. Refcomp from TACO was used to find any overlapping between novel coding/non-coding transcripts and previously known transcripts, identifying intergenic/intronic novel transcripts.

As for orthologous transcripts with human, optimal ORF regions were identified for each of the novel coding transcripts, using Transdecoder (v5.2.0) with default parameters. ORF regions were scanned for homology to verified human peptide sequence (downloaded from Swiss-Prot through http://www.uniprot.org/uniprot/?query=*&fil=reviewed%3Ayes+AND+organism%3A%22Homo+sapiens+%28Human%29+%5B9606%5D%22) via blast (v2.8.0+) and pfam (release 31.0). One-to-one orthologous transcripts between human and rhesus macaque were identified using InParanoid[61] (v4.1) with default settings (score cutoff: 100). Gene symbols of orthologous transcripts in rhesus macaque were decided by converting human protein ID into ENSEMBL transcript and gene IDs according to UniProt.

Region-associated novel transcripts were identified using LRT comparing full model ~Rhesus ID + Region with reduced model ~Rhesus ID (FDR < 0.05).

**lncRNA analysis**. Based on previous analysis of de novo assembly of transcriptome, we selected novel transcripts located in intergenic region or antisense to known genes. The minimal distance between candidate lncRNAs and the nearest protein-coding genes (annotated by Mmul 8.0.1.91) was set to 1 kb. Also, the candidate lncRNAs should be assigned as non-coding by either CPC2 or CPAT. In addition, transcripts shorter than 200 bases and consisted of one single exon were also excluded to reach the final list of lncRNAs.

Co-expression analysis was performed with these lncRNAs using WGCNA (soft-thresholding powers of 3, other parameters same as above). The lncRNA modules were characterized based on the expression of the module eigengene (ME). The Pearson correlation between each region and each ME was calculated, and Student asymptotic $p$-value was calculated for given correlations. Modules were considered highly correlated if correlation coefficient not <0.8 and $p$-value < 0.01.

For lncRNA modules highly correlated with specific brain regions, RBP motifs enrichment analysis was performed using AME (v5.0.1, default parameters) using all lncRNAs as control sequences. Motifs enriched (adjusted $p$-value < 0.1) in at least one of the modules were included in further analysis.

To further explore the conservation of rhesus macaque brain lncRNAs, we downloaded human lncRNAs from the NONCODE[62] ($n = 172,216$), LNCipedia[63] ($n = 107,039$) and GENCODE[64] (v28, $n = 28,468$). Among them, 2845 novel lncRNAs were aligned to human lncRNAs using blast (v2.8.0+).

**ATAC-seq data processing and analysis**. Our pipeline for ATAC-seq data processing could be described as follows (Supplementary Fig. 7a). After removal of adapters (cutadapt v1.18, parameter: -a CTGTCTCTTATACACATCT -A CTGTCTCTTATACACATCT -g AGATGTGTATAAGAGACAG -G AGATGTGTATAAGAGACAG --minimum-length 30 -q 10), paired-end ATAC-seq reads generated from brain samples of rhesus macaque were aligned to the rhesus macaque genome (Mmul 8.0.1, downloaded from ENSEMBL), using bwa mem (v0.7.12) with -M parameter. First, PCR duplicates were removed using Picard (v1.119). Reads mapped to mitochondrial DNA were then excluded. Only uniquely mapped and properly paired reads with insert size <2 kb and mapping quality over 30 were kept for downstream analysis.

ATAC-seq peak calling was performed with Genrich (v0.6, available at https://github.com/jsh58/Genrich, parameters: -r -m 30 -q 0.05 -a 200 -j -y -e MT,Y -b) for each region. ATAC-seq data of the same region from different individuals were treated as replicates in peak calling, to generate peak sets for each region. DiffBind[65] (v2.12.0) was used to merge peak sets of different regions to generate a consensus peak set, which is defined as the union of input peak sets (peaks were merged if they overlapped by at least 1 bp). For each peak in the consensus peaks, if it overlapped with only one of the six region peak sets, then it was labeled as reproducible in one region (reproducible 1), etc.

DiffBind was also used for counting reads in each peak and differential peak analysis between two regions. Differential peaks were defined as abs

(log2FoldChange) > 1 and BH-adjusted FDR < 0.1. As for correlation analysis between open chromatin and gene expression, we first applied TMM normalization (full library size) in DiffBind to normalize the read count matrix of ATAC-seq peaks. We then calculated the Pearson correlation coefficient between the normalized intensity of every ATAC peak (log2(TMM + 1)) and expression of every gene (normalized as VST) within the same chromosomes. Correlated peak-gene pair was defined as peak-gene pair with Pearson correlation coefficient of higher than 0.7 and BH-adjusted FDR < 0.1. The gene-peak distance was defined as the distance between the peak and TSS of the gene within each of the pairs.

The BED files recording ATAC-seq signals were converted to BedGraph using bedtools (genomecov command) and scaled by a factor of 1,000,000/LibrarySize. Then the scaled BedGraph files were converted to BigWig files using bedGraphToBigWig for genome visualization. EnrichedHeatmap[66] (v1.14.0) was used to visualize normalized ATAC-seq signal around the center of differential peaks (3 kb up- and downstream).

ChIPseeker[67] (v1.20.0) was used to annotate the function of peaks. To examined the overlap between the CA1-PCG differential peaks and known enhancers, we used the published cortical and subcortical enhancers in macaque brain[42] on the rheMac3 genome and converted the coordinates to Mmul 8.0.1.91 using UCSC liftOver. We then compared the overlap between the differential peaks with the cortical and subcortical enhancers, and the overlap between consensus peaks and known enhancers, using chi-square test.

Motif enrichment analysis was performed on the differential peak with GimmeMotifs[68] (v0.13.1, default parameters, motif database gimme.vertebrate. v5.0). Enrichment score was defined as −log10(corrected p-value).

**Reporting summary**. Further information on research design is available in the Nature Research Reporting Summary linked to this article.

## Data availability

Sequencing data are available at the GEO under accession GSE128537. The source data underlying Figs. 1e, 2a–c, 3a, 3c–e, 4a, 4d and 5a, b and Supplementary Figs. 1c, d, 2a, c, 3a–c, 4a–h, 5c–e, 6a–c, 7e–f and 8a–d are provided as a Source Data file.

## Code availability

The R source code for data analysis and visualization in this study can be found at https://github.com/KeyingLu/Rhesus-macaque-brain.

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

## Acknowledgements
We thank Lin Xia and Kailing Tu for proofreading the paper. We thank all the members of the Dan Xie Lab for valuable discussions and comments. This work was funded by the National Key R&D Program of China (No. 2016YFC1000601, Y.Y.); National Natural Science Foundation of China (Grants No. 91631111, 31571327, and 31771426, D.X.); National Natural Science Foundation of China (Grant No. 81570101 and 81871162, Y. F.); Major Science and Technology Projects of Yunnan Province (Grant No. 2017ZF028, Y.Y. and T.T.); National Natural Science Foundation of China (Grant No. 81802096, S. Y.); National Natural Science Foundation of China (Grant No. 11671073, W.H.); and Development Plan Project of Jilin Provincial Science and Technology Department (Grant No. 20160520105JH, W.H.).

## Author contributions
Y.F., D.X. and Y.Y. conceived and supervised the study. J.W. procured and dissected all the samples. T.T. and J.T. performed the experiments. S.Y., K.L., X.L., X.H., W.H. and H.W. analyzed the data, and created the figures. S.Y., K.L. and D.X. wrote the paper with input of other authors. All authors revised the paper.

## Competing interests
The authors declare no competing interests.
