## [Peer Review File · Nature Communications]

Reviewers' Comments:

Reviewer #1:

Remarks to the Author:

This is an impressive amount of work showing the generation of 452 RNA-Seq samples representing 52 brain regions across 8 individual monkeys. These appear good quality samples at a whopping average of 60 million reads per sample generated. There's also an additional 29 ATAC samples. The number of different brain regions make this data unique and could be published in nature comm as a resource. I also do not know of any other ATAC-Seq data in rhesus brain. There are no surprises in the analysis which yield no real new biological insight but given the size of the dataset I guess that is ok.

I'm sceptical about the interindividual variability analysis as technical replicates are lacking and only 8 individuals are assessed. Were multiple brain regions used as technical replicates in the ANOVA? There is also an overall disregard for differences in cellular heterogeneity playing a role in differences between brain regions. Recent single cell analysis demonstrated that most cortical sample variation was due to differences in heterogeneity which especially in the cortex can emerge as a result of variation in dissection of the region. The text needs editing to improve clarity. Materials and methods as well as figure legends lack detail and there is much redundancy in figure panels.

Specific comments:

Text:

Line 44 ◊ what is meant with at rest state?

Line 109 ◊ what is mixed composition? Wrong sampling?

Line 114 ◊ Fig 1c should be Fig 1d

Lines 131-132 ◊ no parenthesis around abbreviations

Lines 137 ◊ I don't buy that for a second, it is much more likely that differences in heterogeneity play a role here.

Line 141 ◊ states "identified 10 clusters" but these were pre-set 10 for k-means. Why 10? Was this optimized?

Line 186 ◊ according to Fig S2d, LMTK2 not cortex specific

Line 211 ◊ module A, but this is never defined

Are these similar to the already defined modules or were new modules created with only the cortex samples

Line 224 ◊ what was peculiar about them ?

Line 238 ◊ is that because these regions comprise of glial cells primarily??

Line 243 ◊ header states "individual specific" but genes are expressed in all individuals, and a bit up in some. Individual variations would be better. How big is the effect size?

Line 268 ◊ one specific module for Rm3. Does nicotine GO-term make any sense? Does agonal state of the monkey play a role here ?

Line 281 ◊ For the brain regions: methods states that regions were group in 16 groups and DESeq2 was used to define DE regions young-mid age. Is there a rationale for picking these two age groups? Does it coincide with a particular rhesus developmental time ?

Are the samples merged and the average taken per group, or the max expression or are the raw values taken along.

If latter, how does the cortex variability influence the ability the find DE genes in group cortex between young-mid age

Line 285 ◊ Does not correspond with figures. Rhesus10 == 5y and therefore belongs to mid age, but is always included in young group

Line 302 ◊ is difference in gene expression level due to differences in glial content? Does sample

number play a role ?

Line 305-6 ◊ no way

Line 394 ◊ is CRH expression not observed in hypothalamus or not DE?

Line 443 ◊ "varied expression" not clear from figure. E.g. ENO1, MAD2L, MRPL14 are pretty similar. Different x-limits for all plots makes it hard to compare genes

Line 480 ◊ rhesus 4,6,10,1 are all female so is TU10540 gender specific

Line 583 ◊ how is gene-peak pair defined to calculate correlation?

Methods states: first correlation, 2nd look at distance which is vague

Line 589 ◊ really distinct cluster or are too many clusters defined?

Both gene expression (7d), motif enrichment (7e) and correlation (S11b) are similar for cluster 1 and 2. So is this an artefact of the analysis?

Line 589 ◊ states that clusters 1 & 3 are complementary but figures show similarity between 1 & 2

Line 606 ◊ on which set is the DESeq analysis performed. All mappable peaks or on the DE peaks?

If done on the later, this would explain the low amount of DE region in human

Line 760 ◊ What is elevated temperature

Line 776 ◊ How were cells isolated from the brain?

Information about nuclei isolation (4.5) and grouping of sample (4.8) is missing

Unclear when individual regions are used and when aggregates

E.g. 5g: is the cortex an average/max of all regions or is only 1 region shown. (because there is apparent variability)

Figures:

2f ◊ legend stated barplot but violin plot is shown

2f ◊ Unclear what the colors represent

3a ◊ same plot as 2d, but different coloring

5 ◊ figure panel position is messy. Panel d after e/f/g

5f/g/h ◊ which cortical area is shown?

5d ◊ Unclear what the purpose of this chart is. Especially for outer rims

7b ◊ texts mention 3 clusters; what is the group "different peaks"

S1a/b ◊ not stated which is RNA and which is ATAC. Now it looks like an overall plot for RNA seq. especially because ATAC is not mentioned before fig 7

S1b ◊ should be presented in either S10 or S11 with other ATAC data

S2d ◊ should show cortex specific expression, but the genes are clearly expressed in other tissues as well

S2e ◊ gene names not readable

S3b ◊ not mentioned in the text

S5c ◊ not mentioned in the text, only states are: HIP, CB & STR

S5c/d ◊ why no heatmap as in S5a/b if similar data/message

S6 ◊ different coloring of M/F compared to all other figures (e.g. fig. 4)

S6c/d ◊ seem to be a more adult/young difference than M/F

S7 ◊ legends for panels f and g are missing

S7f ◊ x-axis limits are different per gene so difficult to compare variability between genes.

Reviewer #2:

Remarks to the Author:

The manuscript by Yang Yu and colleagues provides a comprehensive expression atlas of rhesus macaque brain. Extensive analysis supports the quality of the data and sheds light on the pattern of transcriptomic variations across brain regions, age, and gender. Additionally, the authors surveyed chromatin accessibility landscape in several cortex regions, which potentially

complements transcriptome analysis.

Overall, I think this large dataset is a very useful resource for researchers studying brain genomics, especially for those interested in how human-specific brain functions evolved. My main criticism is that the paper is quite long, and sometimes made me feel a lack of focus. It would also be helpful if they can strengthen some of the more interesting findings.

Major comments

(1) I think the paper would be easier to read if the authors make it more concise. For example, the last two paragraphs on 2.1. studies brain-specific gene modules. The next section is also about finding gene modules based on co-expression patterns, with a different tool. Conceptually, these are similar things (though details may differ). I'd suggest to somehow consolidate the two parts (e.g. removing the last part of 2.1).

As another example, 2.2 discusses many co-expression modules. While some of them are clearly interesting, it's really hard for a reader to digest these many results and know what the key message is.

Yet another example, I found that the genes and open chromatin peaks differentially expressed across individuals are harder to interpret (comparing with those differ between regions, gender or age). I'd suggest to shorten those sections (unless they find something new, see below).

(2) Section 2.3. about individual-specific genes: I think how this analysis is done should be described more clearly in main text. There are $(8 \text{ choose } 2) = 28$ pairs of individuals, so there are many differential expression analysis to be done, even with only one region. Does the paper use random effect model? Some description here would help one understand the analysis.

About the results, I would think the top differential expression (DE) genes are those that vary the most across individuals. So we are talking about variability of gene expression. But I would think some genes are just more variable than others, e.g. because they have different promoter architecture, or they are more functionally important in general. This may not have much to do with brain biology. Have the authors compare these DE genes with similar analysis done in other tissues reported in literature?

(3) Section 2.4, age-related difference: Are the p-values in Figure 4 adjusted for multiple testing? The enrichment seems weak (except MO) if there is no adjustment.

Some specific results: the authors reported higher expression of meiosis-related genes in the striatum, and interpret them as "adult neurogenesis". But I'd think you should have mitosis genes in neurogenesis, not meiosis.

In cortex and striatum (Figure 4e and f): young monkey brains are enriched with genes involved in rheumatoid arthritis and asthma. This seems puzzling (if results are statistically significant). Any explanation/hypothesis? Could this be due to the role of immune system in shaping synaptic plasticity during development?

(4) Section 2.6. novel transcript analysis: my understanding is that all the analysis in the second half (starting from line 434) are about coding transcripts. It would be good to clarify. About these novel transcripts, it would be good to provide more evidence that they are actually functional. A simple analysis is to check cross-species conservation of these sequences. Another possible strategy is to re-analyze RNA-seq data of related species to show that some of the novel coding transcripts are also expressed in other species.

(5) Section 2.7. novel lncRNAs: two comments here. First, the authors reported enriched motifs in some lncRNA modules. It's hard to know if this finding is real/important, and I think some

additional analysis would strength the results. Ex. are any of these TFs known to be important in brain development? If the model of lncRNA acting as sponges of TF is correct, we should expect that when lncRNA levels are high, the targets of the TFs would have low exprssion. Can the authors test this? Of course, TF targets may be hard to find, but since they have ATAC-seq data, perhaps they can scan for motifs within ATAC-seq peaks - this would give a reasonable set of TF targets.

Secondly, my impression from literature is that lncRNAs may also inhibit RNA-binding proteins (RBPs), perhaps also throught a sponge-like mechanisms. It would be interesting to test RBP motifs as well.

(6) Section 2.8, comparison with human: one main finding here is that in human, amygdala (AMY) and hippocampus (HIP) samples cluster with cortex samples from the same donors. The authors interpret this as "evolutionary shifting of amygdala and hippocampus to resemble function and cell composition of cerebral cortex". If this is true, this would be a quite significant finding. But the authors fail short of doing more analysis to support this result. I have some specific comments/questions here: first, in Figure 6c, it's really hard to see the clustering pattern reported by the authors (that AMY and HIP cluster by donors). I'd suggest better presentation here. Secondly, the correlation analysis of orthologous genes between human and monkey show very high correlation in AMY and HIP samples. This seems contradictory to what's reported. Any explanation? Finally, is it possible to identify specific genes that drive the difference of clustering patterns between human and monkey? Finding these genes may shed some light on the results they found. Perhaps start with genes differentially expressed between AMY/HIP and other regions. The lists may be quite different between human and rhesus.

(7) Section 2.9, ATAC-seq results: overall, my feeling is that this section doesn't have particularly interesting findings. One possible analysis to do is to compare open chromatin regions between rhesus with human (e.g. from PsychENCODE). This may suggest sequences that underwent evolutionary change during human speciation. Such epigenomic comparison across species has been done before in brain, e.g. see PMID: 25745175.

Minor comments

- Line 267, "With WGCNA on the ,", a missing word here.
- Line 398, "As only about 30% ...", this sentence does not read correct.
- Line 406: What is TACO?
- Line 417: citations about CPAT and CPC2
- Line 505-507: need citations about sponge effect of lncRNAs on TFs
- Figure 1d: monkeys are named from 2-11, rather than 1-8. Seems strange since the paper had only 8 monkeys. Also what do colors mean in this figure? Need legend.

Reviewer #3:

Remarks to the Author:

The authors of this manuscript have generated a fantastic rhesus macaque brain transcriptional and chromatin accessibility dataset that significantly extend current rhesus macaque brain atlases (e.g., Bakken et al, 2016; Zhu et al, 2018). The dataset is quite unique, and the authors should be applauded for their herculean effort in generating these valuable data. However, I do have major concerns with the study design, sequencing approach, statistical analysis, and, consequently, the

interpretation, which I detail below.

1. Overall, the writing is difficult to follow. For instance, line 53-56 could easily be rewritten as "For instance, the nuclear organization of the rhesus macaque hippocampus is more similar to that of humans than rodents." This writing style becomes even more difficult to parse when getting into the technical details in the results.

2. The sample size, in terms of number of individuals ($n=8$), is extremely underpowered for a comparison of age and sex classes within regions. Testing differential expression between two groups (middle aged vs. young; male vs. female) within each region will contain many false positives. While the authors did correct for multiple hypothesis tests (which FDR approach did they use with the $p.adjust$ function?), I would encourage them to run permutations (randomize sex and age) to construct an empirical null distribution. What is the number of genes that we would expect to pass their significance threshold ($\log_2FC > 2$ and $p < 0.05$) by chance? With such a small sample size, this is a very important null to generate and test. I would also encourage the authors to use this helpful R package to see what effect sizes they do have the power to detect: <https://www.bioconductor.org/packages/release/bioc/vignettes/RNASeqPower>

3. I am concerned with the choice of an ANOVA for analysis of differential expression across regions and individuals. Did the authors use a one-way repeated measures ANOVA to control for the fact that they had 8 individuals each repeated 52 times? And these individuals were presumably sampled at different times of day, had different post-mortem intervals, etc (see next comment). What is the justification for using an ANOVA to model the RNAseq counts, rather than using a more appropriate model (e.g., mixed-effects negative binomial or Poisson; doi: 10.1093/nar/gkx204)? The use of ANOVA here seems wholly inappropriate given the fact that RNAseq data violate many of the assumptions of ANOVAs (and linear models) in addition to the fact that repeated measures were not properly accounted for.

4. There is no detail on the amount of sample use from each region of the brain. How large were the sections that were removed (I assume a lot given the fact that $> 3 \mu\text{g}$ of RNA was recovered for each region)? Were biopsy punches used? What landmarks were used for dissecting specific regions? How were the brains stored after euthanasia (were they frozen? I find it hard to believe that the specialist was able to sample all 52 regions from the fresh brains immediately)? What is the post mortem interval for each sample? These details are all missing from the methods. Section 4.2 contains far too little detail, and there is no additional detail in the supplement. Verification of some of the sampling regions with landmarks should be presented to confirm sampling accuracy. This is particularly important in the cortical regions, which might be harder to standardize across individuals.

5. There is not enough detail on the RNA-seq data. How many genes were detectably expressed? What was the filtering cutoff for including the sample in the differential expression analysis? What was the relationship between the effect size of age (or sex) and the expression of a gene? These would all inform the power of the study and comparisons. Further, this information would help inform the low genic mapping rate (30%). While total RNAseq is likely to lead to lower mapping to exonic regions, this number is pretty low for a well-annotated genome like the rhesus macaque — especially given the fact that much of the brain transcriptome has been sequenced and annotated in NCBI. Could the high number of intronic reads be due to the fact that a lot of nuclear RNA is being sequenced, which would contain a lot of pre-mRNA (incompletely spliced reads)? Could there be DNA contamination (leading to the intergenic mapping)? What is the proportion of reads that mapped to rRNA, despite the ribo-depletion? What are the RIN values for each sample and were they controlled for in the modeling? There is a lot of inter-sample and inter-region variation in these scores and that could drastically affected your comparisons across regions and individuals. Without these data it is not possible to trust the results of the differential expression or WGCNA analyses.

6. There were 10 clusters detected in the RNA-seq analysis, which was the maximum number of clusters set in the analysis (L832). The fact that the authors ended up at their upper threshold suggests that the number of clusters is likely more than 10. What happens if the maximum number of clusters was set to a larger number?
7. Section 4.2 is very difficult to follow and lacking in detail. What is the "group" in L827? How many genes had an adjusted p-value < 0.05 ? Why were the top 5000 (or 500 for individual analyses) chosen if there were more than that with strong evidence (albeit from an ANOVA) for region or individual specific expression?
8. The individual-specific results seem to be largely driven by Rhesus3. What is it about that individual that makes their expression profiles so different? What happens if you repeat the analysis without that monkey?
9. What proportion of the ATAC-seq data mapped to the mitochondrial genome? How did this vary across samples and did it co-vary with age or sex?
10. The interpretation is often incomplete. A few examples: L269-279, why is this "intriguing"? L224-225, why is this "peculiar"? L372-373, why is this "unsurprising"?
11. While the data have been made available through SRA, there is no code to recapitulate the analyses. Code for analyses should be posted on github (or something like that) so that others can clearly recreate the analysis and apply it to their data.
12. This is relatively minor, but macaques do not have "gender" (a social construct/self-identification), rather they have "sex" (a biological construct). Thus, the authors should remove all of the instances of the word "gender" from their manuscript and replace it with "sex".
13. This is also minor, but the methods say that young animals were < 4 yrs old and older animals were > 4 yrs old, but the results say < 5 yrs and > 5 yrs. It should be 5 (if the supplementary table is correct). In fact, these data could go in Figure 1 so that the reader doesn't have to dig into the supplement to see what the age and sex distribution is. It's only 8 numbers.

Signed: Noah Snyder-Mackler, University of Washington

The detailed point-by-point responses are provided below. The reviewers' comments are listed below in *italicized* font and specific concern been numbered. Our response is in red text and changes / additions to the manuscript are given in blue text.

Reviewer #1 (*Remarks to the Author*):

This is an impressive amount of work showing the generation of 452 RNA-Seq samples representing 52 brain regions across 8 individual monkeys. These appear good quality samples at a whopping average of 60 million reads per sample generated. There's also an additional 29 ATAC samples. The number of different brain regions make this data unique and could be published in nature comm as a resource. I also do not know of any other ATAC-Seq data in rhesus brain. There are no surprises in the analysis which yield no real new biological insight but given the size of the dataset I guess that is ok.

I'm sceptical about the interindividual variability analysis as technical replicates are lacking and only 8 individuals are assessed. Were multiple brain regions used as technical replicates in the ANOVA?

Response: In the original manuscript, we used multiple cortical regions from the same individual as biological replicates in ANOVA. In the revised paper, we have replaced ANOVA with the more robust likelihood ratio test (LRT) with BH-adjusted FDR correction approach, according to the kind suggestions from the reviewers. The LRT examines two models for the counts, a full model with a certain number of terms and a reduced model, in which some of the terms of the full model are removed. The test determines if the extra terms could better explain the data.

There is also an overall disregard for differences in cellular heterogeneity playing a role in differences between brain regions. Recent single cell analysis demonstrated that most cortical sample variation was due to differences in heterogeneity which especially in the cortex can emerge as a result of variation in dissection of the region.

Response: We thank the reviewer for the suggestion. Due to the anatomical variation of the rhesus brain, there is no fully automated and intervention-free dissection methods available. We took measures to minimize the variation in dissection of the brain regions by adopting standardized sampling process using well-defined anatomical landmarks. To minimize operator variation, all dissection was performed by one professional animal anatomist with thorough knowledge of the rhesus macaque brain structures. We have revised the methods section to include more details on the dissection process.

Line: 541-548

Difference in cellular heterogeneity could play a role in differences between brain regions. Further studies, such as single cell analysis, are needed in the

rhesus macaque brain to support this theory. We have revised the text to present such possibility.

Line: 124-126 “Such clustering patterns suggested that non-cortical structures had less inter-individual heterogeneity than the cerebral cortex, which might be due to the difference in cellular heterogeneity.”

The text needs editing to improve clarity. Materials and methods as well as figure legends lack detail. redundancy in figure panels.

Response: The text of the revised paper has been edited to improve clarity, including the methods and figure legends. The figure panels have also been reorganized, leaving 7 figures and 8 supplementary figures.

Specific comments:

Text:

Line 44 what is meant with at rest state?

Response: The rest state means the rhesus brain was not performing tasks or under specific external stimuli before execution, and the individual was generally healthy. This term is to reflect the applicability of the transcriptional map of the rhesus brain. We removed this term to avoid confusion.

Line 109 what is mixed composition? Wrong sampling?

Response: The midbrain is relatively broad, located between the diencephalon (thalamus, hypothalamus) and pons, which includes the ventral tegmental area (VTA) and substantia nigra (SN). In our study, the midbrain was defined as the remaining tissue between diencephalon and pons, after extraction of VTA and SN. To ensure the sampling accuracy for VTA and SN, only a small sample from the central part of VTA/SN was extracted. Therefore, it is likely that some residual tissue from VTA/SN remained in the midbrain sample. Indeed, we found that most of the midbrain samples clustered with VTA samples on t-SNE plot. As a result, the midbrain samples were excluded in downstream differential analysis. We edited the text to reflect the detailed reason for exclusion of midbrain samples in downstream analysis.

Line (results, 2.1, “due to possible mixture of residual tissue from ventral tegmental area (VTA) and substantia nigra (SN)”): 104-106

Line (methods 4.2): 550-554

Line 114 Fig 1c should be Fig 1d

Response: The text has been corrected.

Lines 131-132 no parenthesis around abbreviations

Response: This part has been removed due to redundancy.

Lines 137 I don't buy that for a second, it is much more likely that differences in heterogeneity play a role here.

Response: We revised the text to “Such clustering patterns suggested that non-cortical structures might have less inter-individual heterogeneity than the cerebral cortex.” for clarity.

Line 141 states “identified 10 clusters” but these were pre-set 10 for k-means. Why 10? Was this optimized?

Response: In the original analysis, k from 2 to 15 were attempted and the relative change in area under CDF curve^[1] was close to zero when k exceeded 10. Therefore k was set to 10. However, this part of analysis has been removed in the revised manuscript due to redundancy.

Line 186 according to Fig S2d, LMTK2 not cortex specific

Response: Although brain regions other than the cortex expressed LMTK2, the cerebral cortex had higher expression of LMTK2. We have revised the text to “LMTK2 (in M9) showed high expression in the cerebral cortex. “ as the more accurate statement.

Line 211 module A, but this is never defined

Are these similar to the already defined modules or were new modules created with only the cortex samples

Response: The module A was created with WGCNA using only the cortex samples. We have renamed the module to “Occipital module” for clarity. The text has also been revised accordingly.

Line: 166

Line 224 *what was peculiar about them ?*

Response: The region-associated expression profile has been discussed in the revised version of section 2.2. The pairwise comparison between regions has been removed due to redundancy.

Line 238 *is that because these regions comprise of glial cells primarily?*

Response: According to previous study on human striatum, the caudate nucleus, putamen and nucleus accumbens have similar glial density and glial-to-neuron ratio of almost two[2] (Fig.3b and 3c). Such difference might be due to difference in glial subtypes, but the evidence is lacking, especially in the rhesus brain. After switching to LRT for DE gene discovery, this section has been revised to focus more on genes with higher region-specificity.

Line 243 header states “individual specific” but genes are expressed in all individuals, and a bit up in some. Individual variations would be better. How big is the effect size?

Response: The term “individual specific genes” has been replaced with “genes with high individual-variation”, thanks to the reviewer’s comment. Since the Rhesus ID was categorical (eight individuals), we are not sure how to quantify the effect size of individual variation on gene expression. To assess and present the significance of individual variation for each gene, we listed the p-values and BH-adjusted p-values (FDR) in the newly added Table S10, which reflected the effect size.

Line 268 one specific module for Rm3. Does nicotine GO-term make any sense? Does agonist state of the monkey play a role here?

Response: We found these functional enrichments puzzling as well. We removed this part to make the manuscript more concise and to avoid possible confusion.

Line 281 For the brain regions: methods states that regions were grouped in 16 groups and DESeq2 was used to define DE regions young-mid age. Is there a rationale for picking these two age groups? Does it coincide with a particular rhesus developmental time?

Response: The cut-off age to separate young (or adolescent) from mid-aged (or adult) remained controversial and arbitrary in previous studies. Cut-off

ages at 44 months^[3], 5 years^[4], 4-8 years^[5] have been proposed, which mostly centered at around 4-5 years. Therefore we selected 5 years to separate our two age groups.

Are the samples merged and the average taken per group, or the max expression or are the raw values taken along. If latter, how does the cortex variability influence the ability the find DE genes in group cortex between young-mid age

Response: In the original analysis, the raw values were used for DE gene analysis between young and mid age rhesus brain, using the default DE analysis function of DESeq2 R package. In the revised manuscript, we used the more robust likelihood ratio test to assess the effect of age and sex. Also, permutation-based FDR correction approach was adopted to minimize the influence of cortex variability noise on DE gene discovery.

Line 285 Does not correspond with figures. Rhesus10 == 5y and therefore belongs to mid age, but is always included in young group

Response: We thank the reviewer for pointing out this mistake in the text. The young group was defined as “not older than 5-years” in data analysis. The text has been corrected to “young (not older than 5-years) and mid-aged (older than 5-years) ”.

Line 302 is difference in gene expression level due to differences in glial content? Does sample number play a role ?

Response: The expression difference identified through bulk-sequencing could be due to difference in glial composition, or expression difference in glial cells. We have revised the text to accommodate this possibility. As for sample number, we have adopted LRT with permutation-based FDR correction, which should minimize the effect of different sample numbers and reduce the risk of false discovery.

Line 305-6 no way

Response: The part has been removed to avoid confusion.

Line 394 is CRH expression not observed in hypothalamus or not DE?

Response: CRH was observed in hypothalamus but not DE between male and female rhesus macaque. In the revised paper, after we switch to likelihood ratio test with permutation-based FDR correction, CRH was no longer among the top sex-differential genes and therefore removed in the revised text.

Line 443 “varied expression” not clear from figure. E.g. *ENO1*, *MAD2L*, *MRPL14* are pretty similar. Different x-limits for all plots makes it hard to compare genes

Response: We revised the text to “Some of these new transcripts had varied expression across different brain regions”. This plot was intended to show varied expression of a single transcript across brain regions, rather than between different genes. We revised the figure and unified x-limit across subplots in Fig.S5g and Fig.S5h for the purpose of visually comparing inter-gene expression.

Line 480 *rhesus 4,6,10,1* are all female so is *TU10540* gender specific

Response: Indeed, female rhesus monkeys had higher expression of *TU10540*, which means *TU10540* might be sex-related. We thank the reviewer for pointing out this and we revised the text to “highly expressed in four female rhesus monkeys” (Line 335) to reflect this finding.

Line 583 how is gene-peak pair defined to calculate correlation?
Methods states: first correlation, 2nd look at distance which is vague

Response: Within each chromosome, we calculated the correlation between every gene and every ATAC peak within the same chromosome. The correlated gene-peak pair is defined according to Pearson correlation (correlation coefficient > 0.7). The distance within each gene-peak pair was used for visual presentation only, not for definition. We have revised the methods (Line 829-834) for clearer description.

Line 589 really distinct cluster or are to many clusters defined?
Both gene expression (7d), motif enrichment (7e) and correlation (S11b) are similar for cluster 1 and 2. So is this an artefact of the analysis?

Line 589 states that clusters 1 & 3 are complementary but figures show similarity between 1 & 2

Response: In the original ATAC-Seq analysis, only three peak clusters were defined and two of which were complementary. The ATAC signals of peak cluster 1 and 2 were similar but not identical nor proportional. We could see clear differences between these two clusters in some individuals (original figure 7d, especially Rhesus3, 7 and 8), and the functional annotation of associated genes also differed. Similarly, the motif enrichment analysis and correlation matrix found similar yet different results between these two peak clusters. We considered these two peak clusters both reflected individual variance in chromatin accessibility, just with different perspective. As for the figures, we intended to show that cluster 1 and 3 were complementary and similarity between cluster 1 and 2 simultaneously. These facts were not contradictory.

Line 606 on which set is the DESeq analysis performed. All mappable peaks or on the DE peaks? If done on the later, this would explain the low amount of DE region in human.

Response: Originally we used DE peaks for DESeq analysis. In the revised paper, we used all mappable peaks for analysis, according to reviewer's suggestion. A total of 16408 reciprocal peak regions were used for DESeq analysis.

Line 760 What is elevated temperature

Response: For intact or partially degraded RNA, RNA fragmentation is required. The elevated temperature means 94°C, which was defined in the Kit protocol from NEB (E7420, available at <https://www.neb.com/protocols/2015/06/09/protocol-for-use-with-purified-mrna-or-ribosome-depleted-rna-e7420>). We have revised the text for accuracy (Line 570).

Line 776 How were cell isolated from the brain?

Response: First, the brain tissue was minced using scalpel into tiny cubes <0.5 mm³ and transferred into a 15mL conical tube (BD Falcon) containing brain tissue digestion medium (8mL pre-warmed hibernate-E medium, 1mg/mL collagenase I and 0.5mg/mL collagenase IV, 1mg/ml papain, 10 U/μl DNase I). We have added more detail regarding cell isolation in Methods section 4.5.

Information about nuclei isolation (4.5) and grouping of sample (4.8) is missing

Unclear when individual regions are used and when aggregates

E.g. 5g: is the cortex a average/max of all regions or is only 1 region shown. (because there is apparent variability)

Response: We have added detailed description about nuclei isolation in Methods section 4.4 of the revised manuscript. As for grouping of samples, individual regions were used. The detailed grouping information could be found in the revised Supplementary Table S1. We have revised the text to clarify this information.

Figures:

2f legend stated barplot but violin plot is shown

2f Unclear what the colors represent

Response: The original 2f was moved to Fig.2e, and was changed to boxplot. The colors represented lobe of origin. The figure legend has been revised accordingly.

3a same plot as 2d, but different coloring

Response: We revised the figure and removed Fig.2d.

5 figure panel position is messy. Panel d after e/f/g

Response: We fixed the panel position for clarity (Revised Fig.5).

5f/g/h which cortical area is shown?

Response: In Fig.5f/g/h, all cortical areas from each individual were shown and overlaid on the same track.

5d Unclear what the purpose of this chart is. Especially for outer rims

Response: The purpose of this chart is to present the genome-wide distribution of novel transcripts. The outer rim showed the detailed distribution, with each dot as a transcript. We have simplified this chart by removing the outer rim according to the reviewer's suggestion.

7b texts mention 3 clusters; what is the group "different peaks"

Response: We fixed the text to "differential peaks" in the revised figures.

S1a/b not stated which is RNA and which is ATAC. Now it looks like an overall plot for RNA seq. especially because ATAC is not mentioned before fig 7

S1b should be presented in either S10 or S11 with other ATAC data

Response: The ATAC data-processing flowchart has been moved to Fig. S7 in the revised figures.

S2d should show cortex specific expression, but the genes are clearly expressed in other tissues as well

Response: We fixed the term from “cortex-specific” to “cortex-related” module. These genes were indeed expressed in various brain regions, but had higher expression in the cortex. We added panels Fig.S2d and S2e to present top hub genes in these two cortex-related modules (M9 and M10).

S2e gene names not readable

S3b not mentioned in the text

S5c not mentions in the text, only states are: HIP, CB & STR

S5c/d why no heatmap as in S5a/b if similar data/message

Response: We removed these panels to make the figures more concise.

S6 different coloring of M/F compared to all other figures (e.g. fig. 4)

Response: We fixed the coloring of M/F in S6

S6c/d seem to be a more adult/young difference than M/F

Response: We have re-analyzed the data, using a more robust LRT with permutation-based FDR correction, considering both age and sex as factors (Line 702-707).

S7 legends for panels f and g are missing

S7f x-axis limits are different per gene so difficult to compare variability between genes.

Response: We revised the figure (originally Fig.S7) and unified x-limit across subplots in Fig.S5g and Fig.S5h for the purpose of visually comparing inter-gene expression. The figure legend has been added.

Reviewer #2 (Remarks to the Author):

The manuscript by Yang Yu and colleagues provides a comprehensive expression atlas of rhesus macaque brain. Extensive analysis supports the quality of the data and sheds light on the pattern of transcriptomic variations across brain regions, age, and gender. Additionally, the authors surveyed chromatin accessibility landscape in several cortex regions, which potentially complements transcriptome analysis.

Overall, I think this large dataset is a very useful resource for researchers studying brain genomics, especially for those interested in how human-specific brain functions evolved. My main criticism is that the paper is quite long, and sometimes made me feel a lack of focus. It would also be helpful if they can strengthen some of the more interesting findings.

Response: We have revised the text and figures thoroughly according to the reviewers. The text has been shortened and integrated. Figures with less importance and relevance have been removed, and some findings were strengthened to make the paper shorter while more focused.

Major comments

(1) I think the paper would be easier to read if the authors make it more concise. For example, the last two paragraphs on 2.1. studies brain-specific gene modules. The next section is also about finding gene modules based on co-expression patterns, with a different tool. Conceptually, these are similar things (though details may differ). I'd suggest to somehow consolidate the two parts (e.g. removing the last part of 2.1).

Response: The gene cluster analysis in the Results section 2.1 has been integrated with WGCNA module analysis for consolidation and section 2.1 has been shortened considerably. The rest of the text has also been shortened to make the manuscript more focused and concise.

As another example, 2.2 discusses many co-expression modules. While some of them are clearly interesting, it's really hard for a reader to digest these many results and know what the key message is.

Response: Discussion of co-expression modules has been shortened and consolidated. The key message (how genes in these region-related modules fit in context of previous research) has been emphasized in the revised manuscript.

Yet another example, I found that the genes and open chromatin peaks differentially expressed across individuals are harder to interpret (comparing with those differ between regions, gender or age). I'd suggest to shorten those sections (unless they find something new, see below).

Response: We found these individually varied open chromatin peaks and associated genes difficult to interpret as well. We have shortened these sections according to the reviewer's suggestion.

(2) Section 2.3. about individual-specific genes: I think how this analysis is done should be described more clearly in main text. There are $(8 \text{ choose } 2) = 28$ pairs of individuals, so there are many differential expression analysis to be done, even with only one region. Does the paper use random effect model? Some description here would help one understand the analysis.

Response: In the original analysis, we used ANOVA to assess the inter-individual variation of gene expression. We did not perform 28 pairwise comparison to reveal individual-associated genes and we did not used random effect model. We have re-analyzed the data using more robust LRT (negative binomial model) with BH FDR correction for identification of genes

with high inter-individual variance. We have revised the methods section with more details to help readers understand the analysis. (Line 684-688)

About the results, I would think the top differential expression (DE) genes are those that vary the most across individuals. So we are talking about variability of gene expression. But I would think some genes are just more variable than others, e.g. because they have different promoter architecture, or they are more functionally important in general. This may not have much to do with brain biology. Have the authors compare these DE genes with similar analysis done in other tissues reported in literature?

Response: We downloaded the GTEx expression matrix for gene variability analysis. The brain samples were excluded. The top 15 tissues with the most individuals were used for this analysis. A total of 10 individuals and 227 samples were included, with complete expression data of these 15 tissues in each individual. LRT was then used to compare full model \sim individual + tissue with reduced model \sim tissue, which ended up with 18343 genes with high inter-individual variability (FDR<0.05). The top 1000 genes with high inter-individual variability in rhesus cortex and human tissues had only 93 genes in common. We have

(3) Section 2.4, age-related difference: Are the p-values in Figure 4 adjusted for multiple testing? The enrichment seems weak (except MO) if there is no adjustment.

Response: For functional enrichment, we used the g:Profiler webserver[6], and the default multiple testing correction method (g:SCS) was used to correct the p-values to reduce false positive findings, with 0.05 as cut-off for adjusted p-value. Only significantly enriched term after correction were returned by g:Profiler and visualized as barplot in our study. We have added this information at the end of Methods 4.9. (Line 716-720)

Some specific results: the authors reported higher expression of meiosis-related genes in the striatum, and interpret them as "adult neurogenesis". But I'd think you should have mitosis genes in neurogenesis, not meiosis.

Response: We found it difficult to interpret meiosis-related gene enrichment as well. In the revised paper, after we switched to LRT with permutation-based FDR correction, this term was no longer within the top enriched terms and thus removed in the text.

In cortex and striatum (Figure 4e and f): young monkey brains are enriched with genes involved in rheumatoid arthritis and asthma. This seems puzzling (if results are statistically significant). Any explanation/hypothesis? Could this be due to the role of immune system in shaping synaptic plasticity during development?

Response: We find these functional enrichments puzzling as well. We agree that this could be explained by the role of immune system in shaping synaptic plasticity during neurodevelopment. We thank the reviewer for valuable input. We have revised the text to reflect this theory (Line 244-245).

(4) Section 2.6. novel transcript analysis: my understanding is that all the analysis in the second half (starting from line 434) are about coding transcripts. It would be good to clarify. About these novel transcripts, it would be good to provide more evidence that they are actually functional. A simple analysis is to check cross-species conservation of these sequences. Another possible strategy is to re-analyze RNA-seq data of related species to show that some of the novel coding transcripts are also expressed in other species.

Response: We revised the text to emphasize that this part was about coding transcripts (Line 294). We also re-analyzed the RNA-seq data to check cross-species conservation of these sequences. We downloaded conservation scores from ensembl website (54_amniotes_gerp_conservation_scores.macaca_mulatta.Mmul_8.0.1.bw and 91_mammals_gerp_conservation_scores.macaca_mulatta.Mmul_8.0.1.bw). In these records, positive scores represent highly-conserved positions while negative scores represent highly-variable positions. The bigWigAverageOverBed (v2) was used to compute the average score of each transcript. We found that most of the predicted non-coding transcripts had close to zero or negative conservation scores, indicating low conservation across the species (see Figure below). Meanwhile nearly half of the predicted coding sequence had positive conservation scores. These results suggested that the predicted coding sequences were more conserved across the species and likely to be functional. We have included these analysis in the supplementary information to help the readers better understand the results.

(5) Section 2.7. novel lncRNAs: two comments here. First, the authors reported enriched motifs in some lncRNA modules. It's hard to know if this finding is real/important, and I think some additional analysis would strength the results. Ex. are any of these TFs known to be important in brain development? If the model of lncRNA acting as sponges of TF is correct, we

should expect that when lncRNA levels are high, the targets of the TFs would have low expression. Can the authors test this? Of course, TF targets may be hard to find, but since they have ATAC-seq data, perhaps they can scan for motifs within ATAC-seq peaks - this would give a reasonable set of TF targets.

Secondly, my impression from literature is that lncRNAs may also inhibit RNA-binding proteins (RBPs), perhaps also through a sponge-like mechanism. It would be interesting to test RBP motifs as well.

Response: This has also been asked by other reviewer. We have replaced the TF-binding motif enrichment analysis with the more appropriate RBP-binding motif enrichment analysis, thanks to the reviewers' suggestions. We attempted to search for the association between lncRNA-enriched TF-binding sites and ATAC-seq peaks, but we did not find meaningful results. With RBP-binding motif enrichment analysis, we scanned the lncRNA sequences in each of the lncRNA modules for enrichment of known RBP-binding motifs and discovered module-specific motifs (Fig. S6c). For instance, the cerebellum-associated lncRNA module were enriched in motifs for RBM38 and ESRP2, while the cerebral-specific lncRNAs were enriched in binding sites for PTBP1, which is a repressor of the neural-specific splicing program and crucial for neuronal differentiation.

(6) Section 2.8, comparison with human: one main finding here is that in human, amygdala (AMY) and hippocampus (HIP) samples cluster with cortex samples from the same donors. The authors interpret this as "evolutionary shifting of amygdala and hippocampus to resemble function and cell composition of cerebral cortex". If this is true, this would be a quite significant finding. But the authors fail short of doing more analysis to support this result.

Response: We compared the distance between amygdala/hippocampus and other regions (cerebral cortex and cerebellum) within each individual. We found that the distance between amygdala and cerebral cortex was significantly smaller in human than rhesus macaque, as is the distance between hippocampus and cortex (both $p.value < 1e-13$, revised Fig. 6c). As reference, the difference between amygdala-cerebellum/hippocampus-cerebellum was insignificant. We have revised the text to include this extra analysis to support our interpretation.

I have some specific comments/questions here: first, in Figure 6c, it's really hard to see the clustering pattern reported by the authors (that AMY and HIP cluster by donors). I'd suggest better presentation here.

Response: We revised the original figure 6c (currently figure 6a) by adding circles to highlight cortex from each individual human (HSB12X), as well as circles that marked clustering of STR, THA and CBC for better and clearer presentation of the clustering pattern on t-SNE.

Secondly, the correlation analysis of orthologous genes between human and monkey show very high correlation in AMY and HIP samples. This seems contradictory to what's reported. Any explanation?

Response: Previous comparison studies between human and rhesus monkey/chimpanzee were mostly based on poly-A RNA-seq[7–9] which focused on mature mRNA in the brain. Meanwhile, it has been reported that frontal cortex of human brain had significantly higher proportion of immature transcripts than liver[10]. It might be due to the difference in function between brain and other tissue and the immature transcripts might accounted for the difference between our study and previously reported results. We have added this point as discussion in the text (Line 462-468).

Finally, is it possible to identify specific genes that drive the difference of clustering patterns between human and monkey? Finding these genes may shed some light on the results they found. Perhaps start with genes differentially expressed between AMY/HIP and other regions. The lists may be quite different between human and rhesus.

Response: We have thoroughly compared the AMY/HIP with other regions in human and rhesus macaque respectively, as reviewer 2 suggested. We found that only a small portion of the differential genes were in common between human and rhesus macaque (see figure below). We removed it from the revised manuscript to shorten the text and to avoid confusion.

(7) Section 2.9, ATAC-seq results: overall, my feeling is that this section doesn't have particularly interesting findings. One possible analysis to do is to compare open chromatin regions between rhesus with human (e.g. from PsychENCODE). This may suggest sequences that underwent evolutionary

change during human speciation. Such epigenomic comparison across species has done before in brain, e.g. see PMID: 25745175.

Response: In the ATAC-seq analysis, we did find individual-associated open chromatin peak clusters with varied functional annotation (Figure 7b), and enrichment of TF-binding motifs (Figure S7e). As suggested by reviewers, we attempted to compare ATAC-seq data in rhesus brain with human brain ATAC-seq data. Reciprocal liftover was used to identify ATAC-seq peaks' orthologous regions. Among a total of 25842 Rhesus peaks, 16408 conserved regions (reciprocal liftover) were found in human genome. We then downloaded human brain ATAC-seq data from GEO (GSE96949)[11]. We found that over 50% of the conserved regions had no overlap with human ATAC-seq peaks in any human sample. We also repeated the comparison with human neuronal/non-neuronal ATAC-seq dataset with all conserved peaks using LRT and found no significant difference. Due to the difficulty in comparing different datasets with varied experimental protocol/laboratory settings, we think it might be better to shorten this part as suggested, to make the manuscript more concise. (Figure below, left: barplot showing the percentage of the number of samples in which a given conservative region was observed as peak; right: venn plot showing the intersect of individual-variant peaks between human and rhesus.)

Minor comments

- Line 267, "With WGCNA on the ,", a missing word here.

Response: This section was removed in the revised text to make the manuscript more concise.

- Line 398, "As only about 30% ...", this sentence does not read correct.

Response: We revised the text to "About 30%" (line 270)

- Line 406: What is TACO?

Response: TACO (Transcriptome Assemblies Combined into One) is a transcriptome assembler that could reconstruct full-length transcripts from the

sort sequence fragments generated by RNA-seq. We have revised the text as below to help readers understand and added citation for TACO.

“Therefore, we next assembled the unannotated transcripts for each brain region using TACO (Transcriptome Assemblies Combined into One, v0.7.3).” (Line 273-275)

- Line 417: citations about CPAT and CPC2

- Line 505-507: need citations about sponge effect of lncRNAs on TFs

Response: We have added citations in the text to help the readers understand the context, thanks to the reviewer’s suggestion. (Line 279)

- Figure 1d: monkeys are named from 2-11, rather than 1-8. Seems strange since the paper had only 8 monkeys. Also what do colors mean in this figure? Need legend.

Response: We have renamed the monkeys from 1 to 8 to avoid confusion. The colors in the original figure 1d means brain regions. This panel has been moved to the revised figure 1e, and we added color legend aside this panel to help readers understand.

Reviewer #3 (Remarks to the Author):

The authors of this manuscript have generated a fantastic rhesus macaque brain transcriptional and chromatin accessibility dataset that significantly extend current rhesus macaque brain atlases (e.g., Bakken et al, 2016; Zhu et al, 2018). The dataset is quite unique, and the authors should be applauded for their herculean effort in generating these valuable data. However, I do have major concerns with the study design, sequencing approach, statistical analysis, and, consequently, the interpretation, which I detail below.

1. Overall, the writing is difficult to follow. For instance, line 53-56 could easily be rewritten as “For instance, the nuclear organization of the rhesus macaque hippocampus is more similar to that of humans than rodents.” This writing style becomes even more difficult to parse when getting into the technical details in the results.

Response: We have thoroughly rewritten the manuscript to make it more concise and focused, thanks to the reviewers’ suggestions. The text and figures have been consolidated considerably to help the readers understand the key message of this paper.

2. The sample size, in terms of number of individuals (n=8), is extremely underpowered for a comparison of age and sex classes within regions. Testing differential expression between two groups (middle aged vs. young; male vs. female) within each region will contain many false positives. While the authors did correct for multiple hypothesis tests (which FDR approach did they use with the p.adjust function?), I would encourage them to run permutations (randomize sex and age) to construct an empirical null

distribution. What is the number of genes that we would expect to pass their significance threshold ($\log_2FC > 2$ and $p < 0.05$) by chance? With such a small sample size, this is a very important null to generate and test. I would also encourage the authors to use this helpful R package to see what effect sizes they do have the power to detect:

<https://www.bioconductor.org/packages/release/bioc/vignettes/RNASeqPower>

Response: In the original analysis, we used Benjamini-Hochberg (BH) approach to correct for multiple hypothesis tests (using $p.adjust$ in R). We thank the reviewer for offering a more robust FDR correction approach. In the revised paper, we re-analyzed the data using likelihood ratio test (nbinomLRT, in DESeq2 R package, based on negative binomial distribution). For age and sex-related DE gene analysis, permutation-based FDR correction was performed. Sex and age of the individuals were randomized for 20 times. In each permutation, LRT was performed to calculate p-value. Permutation-based FDR approach was then used to compare the observed p-value against the empirical null distribution using $fdrci$ (R-package, v2.1), and the number of genes expected to surpass the significance threshold was calculated based on the null distribution. We revised the methods section to reflect this change as below. We also calculated the effect size of age and sex for each DE gene and listed in supplementary Table S11-12, to help the readers understand.

(Methods 4.9, line 708-714)

3. I am concerned with the choice of an ANOVA for analysis of differential expression across regions and individuals. Did the authors use a one-way repeated measures ANOVA to control for the fact that they had 8 individuals each repeated 52 times? And these individuals were presumably sampled at different times of day, had different post-mortem intervals, etc (see next comment). What is the justification for using an ANOVA to model the RNAseq counts, rather than using a more appropriate model (e.g., mixed-effects negative binomial or Poisson; doi: 10.1093/nar/gkx204)? The use of ANOVA here seems wholly inappropriate given the fact that RNAseq data violate many of the assumptions of ANOVAs (and linear models) in addition to the fact that repeated measures were not properly accounted for.

Response: In the original analysis, we transformed the RNA-seq count data into VST values before performing ANOVA, using the embedded function within the DESeq2 R package. After transformation, our data did not violate those assumptions of ANOVA/linear models. We agree that negative binomial model is more appropriate for RNA-seq count data. Therefore, we re-analyzed our data using likelihood ratio test (nbinomLRT, in DESeq2 R package, based on negative binomial distribution) which considered multiple factors simultaneously within the models. In addition, we used the more robust permutation-based FDR correction approach according to the reviewer's suggestion. As a result, we were able to identify differentially expressed genes with more confidence. Also, the new top DE genes have been thoroughly

discussed to provide more biological insight as well as research context (Results 2.4).

4. There is no detail on the amount of sample use from each region of the brain. How large were the sections that were removed (I assume a lot given the fact that > 3 µg of RNA was recovered for each region)? Were biopsy punches used? What landmarks were used for dissecting specific regions? How were the brains stored after euthanasia (were they frozen? I find it hard to believe that the specialist was able to sample all 52 regions from the fresh brains immediately)? What is the post mortem interval for each sample? These details are all missing from the methods. Section 4.2 contains far too little detail, and there is no additional detail in the supplement. Verification of some of the sampling regions with landmarks should be presented to confirm sampling accuracy. This is particularly important in the cortical regions, which might be harder to standardize across individuals.

Response: We added more details of brain dissection and sample processing in Methods 4.1 and 4.2. For each region, about 2 to 3mm³ brain tissue were removed using scalpel and microscopic scissors to make sure that more than 3µg of RNA could be extracted for experiment. We did not use biopsy punches for sampling. We used well-recognized landmarks reported in previous comprehensive anatomical researches on rhesus macaque brain, including the rhesus monkey brain atlas (A Combined MRI and Histology Atlas of the Rhesus Monkey Brain in Stereotaxic Coordinates 2nd Edition) and the Brain Maps (available at [http://www. Brainmaps.org](http://www.Brainmaps.org)). The brain was harvested as soon as the rhesus monkey was euthanized to minimize the ischemic time. The brain was then cooled to 4°C and carefully dissected. The dissected samples were stored in liquid nitrogen before RNA extraction. The time used to harvest each region was between 30 to 60 seconds, and dissection of 52 brain regions from one brain was completed in about 30 minutes.

5. There is not enough detail on the RNA-seq data. How many genes were detectably expressed? What was the filtering cutoff for including the sample in the differential expression analysis?

Response: We have revised the text to include more details on the RNA-seq data according to the reviewer's suggestion. In the revised analysis, a total of 23651 genes were detected (read count>1 in at least 3 samples).

What was the relationship between the effect size of age (or sex) and the expression of a gene? These would all inform the power of the study and comparisons.

Response: Other reviewers also asked about the effect size of age/sex on expression of genes. Thanks to the reviewers' suggestion, we re-analyzed the data using likelihood ratio test. For age and sex-related DE gene analysis, a more robust permutation-based FDR correction was performed as suggested.

We also calculated the effect size of age and sex for each DE gene and listed in supplementary table, to help the readers to understand.

Further, this information would help inform the low genic mapping rate (30%). While total RNAseq is likely to lead to lower mapping to exonic regions, this number is pretty low for a well-annotated genome like the rhesus macaque — especially given the fact that much of the brain transcriptome has been sequenced and annotated in NCBI. Could the high number of intronic reads be due to the fact that a lot of nuclear RNA is being sequenced, which would contain a lot of pre-mRNA (incompletely spliced reads)? Could there be DNA contamination (leading to the intergenic mapping)?

Response: We used Ribo-Zero total-RNA sequencing and this method is known to capture a large proportion of non-polyadenylated RNA[12,13]. Previous report of RNA-seq using rRNA depletion method on human and chimpanzee brain[10] also showed similar intronic mapping rate of ~30% and intergenic mapping rate of ~40%, which is comparable to our result in the rhesus macaque brain.

In terms of immature transcripts (pre-mRNA), report by Ameur et al[10] did suggest that frontal cortex of human brain had significantly higher proportion of immature transcripts than liver. It might be due to the difference in function between brain and other tissue and the immature transcripts might be interesting to study in the brain. Since our data were generally comparable to previous reports in human and chimpanzee brains, the possibility of DNA contamination is low. Besides, we applied RQ1 DNase before RNA-seq library preparation according to Illumina protocol to minimize DNA contamination. We have revised the RNA extraction and RNA-seq library preparation section in the methods to include more details and to avoid confusion.

What is the proportion of reads that mapped to rRNA, despite the ribo-depletion?

Response: About 1% of the reads were mapped to rRNA (ranged from 0.57%~1.35%), indicating sufficient ribo-depletion. We added this data in the revised text.

What are the RIN values for each sample and were they controlled for in the modeling? There is a lot of inter-sample and inter-region variation in these scores and that could drastically affected your comparisons across regions and individuals. Without these data it is not possible to trust the results of the differential expression or WGCNA analyses.

Response: We have added the RIN values for each samples in Table S3. The Illumina Ribo-Zero technique has good performance when sequencing lower quality samples compared with poly-A+ method, including samples with RIN less than 3[13]. In our 418 samples, only two samples had an RNA integrity number (RIN) of less than 5, both with high median transcript integrity number (medTIN >70, see methods, Fig. S1c). The number of expressed

genes ($\log_2(\text{RPKM}+1) > 1$ or 10) in each sample had no correlation with RIN of the sample (Fig. S1d). As correlation between RIN and expression data was weak, especially in Ribo-Zero RNA-seq, we did not use RIN as factor in modeling.

6. There were 10 clusters detected in the RNA-seq analysis, which was the maximum number of clusters set in the analysis (L832). The fact that the authors ended up at their upper threshold suggests that the number of clusters is likely more than 10. What happens if the maximum number of clusters was set to a larger number?

Response: Another reviewer also asked about this. In the original analysis, k from 2 to 15 were attempted and the incremental change in area under CDF curve (see below) was close to zero when k exceeded 10. Therefore k was set to 10. This part of analysis has been removed in the revised manuscript due to redundancy.

7. Section 4.2 is very difficult to follow and lacking in detail. What is the “group” in L827? How many genes had an adjusted p -value < 0.05 ? Why were the top 5000 (or 500 for individual analyses) chosen if there were more than that with strong evidence (albeit from an ANOVA) for region or individual specific expression?

Response: The group referred to group used in ANOVA analysis, and a total of 17411 genes had an adjusted p -value < 0.05 . We chose top 5000/500 genes to best represent genes with the highest region- or individual- specificity. The region-associated gene expression analysis using ANOVA has been removed and consolidated with WGCNA analysis, thanks to the reviewer’s

suggestion. Meanwhile, the individual-associated gene expression analysis in the cortex has been replaced with the more robust and appropriate likelihood ratio test, which is also easier to understand. The methods sections have been revised accordingly to improve readability and provide more details on these analysis, allowing reproduction of our results.

8. The individual-specific results seem to be largely driven by Rhesus3. What is it about that individual that makes their expression profiles so different? What happens if you repeat the analysis without that monkey?

Response: We added an extra expression filtration process (raw count number higher than 1 in at least 3 samples) before downstream analysis according to the reviewers' suggestion. After filtration, we performed WGCNA on cortex samples with Rhesus3 included (currently renamed to Rhesus2 as suggested) and did not find individual-associated gene module (see figure below left). After further removal of this individual, we found multiple individual-associated modules (see figure below right). Considering that the high variation in the cortex might affect the stability of WGCNA, we removed this part of analysis to make the manuscript more focused and to avoid possible confusion.

9. What proportion of the ATAC-seq data mapped to the mitochondrial genome? How did this vary across samples and did it co-vary with age or sex?

Response: Only 1.5%~ 9.8% (mean 4.7%) of the ATAC-seq reads could be mapped to the mitochondrial genome (MT), which is comparable to previous ATAC-seq studies[14,15]. We compared the mitochondrial mapping ratio

between female/male and mid/young and found no significant difference between age and sex groups.

10. *The interpretation is often incomplete. A few examples: L269-279, why is this “intriguing”? L224-225, why is this “peculiar”? L372-373, why is this “unsurprising”?*

Response: We apologize for not elaborating the interpretation enough. These lines have been removed in the revised manuscript to make the text more focused and easier to understand, according to reviewers’ suggestions. The interpretation of other findings in the text has been revised to help the readers understand, even without neurological background.

11. *While the data have been made available through SRA, there is no code to recapitulate the analyses. Code for analyses should be posted on github (or something like that) so that others can clearly recreate the analysis and apply it to their data.*

Response: We also think reproducibility is important for scientific researches. The R source code for data analysis and visualization in this study has been uploaded to Github (<https://github.com/KeyingLu/Rhesus-macaque-brain>) and will be released along with SRA data, after publication of this paper.

12. *This is relatively minor, but macaques do not have “gender” (a social construct/self-identification), rather they have “sex” (a biological construct). Thus, the authors should remove all of the instances of the word “gender” from their manuscript and replace it with “sex”.*

Response: We thank the reviewer for this valuable suggestion. We have replaced the all “gender” with the more appropriate word “sex” in the text and figures as well.

13. *This is also minor, but the methods say that young animals were <4 yrs old and older animals were >4 yrs old, but the results say < 5 yrs and > 5 yrs. It should be 5 (if the supplementary table is correct). In fact, these data could*

go in Figure 1 so that the reader doesn't have to dig into the supplement to see what the age and sex distribution is. It's only 8 numbers.

Response: We thank the reviewers for pointing out this. The young group was defined as “not older than 5-years” in data analysis. The text has been corrected to “young (not older than 5-years) and mid-aged (older than 5-years)”. We have also added the detailed age information of the eight individuals in Figure 1a.

References

1. Monti S, Tamayo P, Mesirov J, Golub T. Consensus Clustering: A Resampling-Based Method for Class Discovery and Visualization of Gene Expression Microarray Data. *Mach Learn.* 2003;52: 91–118.
2. Hanson KL, Lew CH, Hrvoj-Mihic B, Groeniger KM, Halgren E, Bellugi U, et al. Increased glia density in the caudate nucleus in williams syndrome: Implications for frontostriatal dysfunction in autism. *Dev Neurobiol.* 2018;78: 531–545.
3. Andrade MCR, Ribeiro CT, Silva VF da, Molinaro EM, Gonçalves MAB, Marques MAP, et al. Biologic data of *Macaca mulatta*, *Macaca fascicularis*, and *Saimiri sciureus* used for research at the Fiocruz primate center. *Mem Inst Oswaldo Cruz.* 2004;99: 581–589.
4. Simmons HA. Age-Associated Pathology in Rhesus Macaques (*Macaca mulatta*). *Vet Pathol.* 2016;53: 399–416.
5. von Borell C, Kulik L, Widdig A. Growing into the self: the development of personality in rhesus macaques. *Anim Behav.* 2016;122: 183–195.
6. Reimand J, Arak T, Adler P, Kolberg L, Reisberg S, Peterson H, et al. g:Profiler—a web server for functional interpretation of gene lists (2016 update). *Nucleic Acids Res.* 2016;44: W83–9.
7. Sousa AMM, Zhu Y, Raghanti MA, Kitchen RR, Onorati M, Tebbenkamp ATN, et al. Molecular and cellular reorganization of neural circuits in the human lineage. *Science.* 2017;358: 1027–1032.
8. Zhu Y, Sousa AMM, Gao T, Skarica M, Li M, Santpere G, et al. Spatiotemporal transcriptomic divergence across human and macaque brain development. *Science.* 2018;362. doi:10.1126/science.aat8077
9. Xu C, Li Q, Efimova O, He L, Tatsumoto S, Stepanova V, et al. Human-specific features of spatial gene expression and regulation in eight brain regions. *Genome Res.* 2018;28: 1097–1110.
10. Ameer A, Zaghlool A, Halvardson J, Wetterbom A, Gyllensten U, Cavelier

L, et al. Total RNA sequencing reveals nascent transcription and widespread co-transcriptional splicing in the human brain. *Nat Struct Mol Biol.* 2011;18: 1435–1440.

11. Fullard JF, Hauberg ME, Bendl J, Egervari G, Cirnaru M-D, Reach SM, et al. An atlas of chromatin accessibility in the adult human brain. *Genome Res.* 2018;28: 1243–1252.
12. SEQC/MAQC-III Consortium. A comprehensive assessment of RNA-seq accuracy, reproducibility and information content by the Sequencing Quality Control Consortium. *Nat Biotechnol.* 2014;32: 903–914.
13. Li S, Tighe SW, Nicolet CM, Grove D, Levy S, Farmerie W, et al. Multi-platform assessment of transcriptome profiling using RNA-seq in the ABRF next-generation sequencing study. *Nat Biotechnol.* 2014;32: 915–925.
14. Montefiori L, Hernandez L, Zhang Z, Gilad Y, Ober C, Crawford G, et al. Reducing mitochondrial reads in ATAC-seq using CRISPR/Cas9. *Sci Rep.* 2017;7: 2451.
15. Rickner HD, Niu S-Y, Cheng CS. ATAC-seq Assay with Low Mitochondrial DNA Contamination from Primary Human CD4+ T Lymphocytes. *J Vis Exp.* 2019; doi:10.3791/59120

Reviewers' Comments:

Reviewer #1:

Remarks to the Author:

I have no major additional comments on this manuscript but several suggestions.

The writing and presentation has improved substantially but the story still lacks structure. Sections summing up the relationship between expression differences and known brain specific genes are interlaced with sections about interindividual variability (line 190 and 332) and sections about new transcripts, sex and age. It would IMO be better to begin sections about finding known genes related to brain structures (more concisely because its known) as validation of the study samples. Then introduce the new genes showing that new information can arise from the data and then end with interindividual variability (controlling for sex and age) ending with a discussion on what this data may mean. A question that could arise is if the new genes found here have not been found before because there's higher interindividual variability for them. There's also no reason to discuss sex and aging separately but as for controls to the observed interindividual variability. Focusing on these things as separate issues gives the manuscript an unstructured feel and leave the authors open to questions about power.

P222 I'm skeptical the authors can measure microglial cells given how many of the cells within their samples are actually microglia.

The ATAC seq data line p390 is still not well integrated with the RNA-Seq data (do the two correlate?) is interindividual variation in ATAC-Seq correlated with inter individual variation in RNA-Seq. This data now appears as a sideshow (oh we also have this data) to the rest of the manuscript. It could be discussed as part of the transcriptional inter-individual variability part. There's still several spelling and sentence structure mishaps, that force me to infer what the authors mean. The authors could let the manuscript be proofread by a professional organization such as lifescienceeditors or something alike? Naturally that part is only a suggestion.

Reviewer #2:

Remarks to the Author:

The authors have addressed most of my comments. My only remaining questions are about Section 2.3, Discovering genes with high individual-variation:

1) It is unclear how analysis is done. I understand that the authors fit two models, and compare them by LRT. However, in the full model: $\text{expression} \sim \text{individual ID} + \text{region}$, how is "individual ID" coded? If it is binary coded (one variable per sample), then the full model has 8 more variables (hence 8 more coefficients) than the reduced model. It is a bit strange to analyze data in this way, as the two models differ a lot in complexity.

2) In the comparison of two models, I would think age and sex could be major confounders, as they could have large impact on gene expression. So it's possible that the results are largely driven by the extent by which gene expression responds to age/sex, instead of "intrinsic" variability of gene expression. A simple thing to do is perhaps to adjust for age/sex in the regression model.

Reviewer #3:

Remarks to the Author:

This resubmission still represents a fantastic resource that the authors have put together. As far as I know, it is the most comprehensive genomic atlas of a rhesus brain to date. The authors have thoughtfully revised the manuscript in response to my (and others') comments. However, I still have some major concerns, particularly with the statistical analysis, that the authors should address.

major comments:

- I agree with reviewer 1 that this would be a great publication as a resource in Nat Comm (or another journal that publishes resources), but there isn't much new biological insight/interest. However, I'll leave this decision up to the editor.

- L124-126: cellular heterogeneity is clearly something the authors would like to highlight here. Could the authors test this hypothesis by doing a deconvolution analysis using the single-cell data available in Zhu et al, 2018, Science, to test their potential interpretation here?

- Section 2.3. There are two reasons why the authors cannot say much about individual differences, and why I would suggest they remove this whole section and most interpretation:
1. The authors are still severely underpowered to test for inter-individual differences and the LRT approach doesn't help with an n=8
2. There are many technical reasons for differences among individuals. For instance, the animals were sampled on different days, perhaps libraries prepped on different days, RNA extracted on different days, slight variation in the PMI, etc. For this reason, I don't think it is really valid to interpret this as biological differences among individuals without clearly acknowledging that these might be technical or without significantly increasing the sample size.

Further, if I am interpreting the authors correctly, then they are using a likelihood ratio test to see if this model:

~ individual + tissue

fits better for a gene than this model:

~ tissue

I think this is an inappropriate way to analyze the data for the reasons I outlined above (and because sex and age aren't included). There are many technical, rather than biological reasons, for this to be a better fit. The authors also omitted age and sex from this model. In sum, I would encourage the authors to remove the sections on individual differences. It's not necessary for the resource.

- Section 2.4: Thank you for including the details of the LRT in the methods (section 4.9). Having seen these details I now see that the authors are inappropriately assuming that the repeated samples from individuals are independent and not correlated with age and sex (which they are!). This makes it very difficult to trust their findings.

Further, they are assuming that all of the regions within the 16 "closely related sections" are all the same. Am I correct in reading that the authors are comparing these two models?

~ Age + Sex

vs

~ Age (or ~Sex)

if so, then the authors needs to change the model to compare:

~ Age + Sex + Individual + Region

vs

~ Age + Individual + Region (or ~ Sex + Individual + Region)

Otherwise they are pseudoreplicating their data and age/sex effects (which could be technical) could be misinterpreted as individual differences. It would be even better if the authors could include individual ID as a random effect in a mixed-effects model framework.

- L709-710: did you permute ages within individuals or across them? In other words, for each permutation, did different regions from the same individual get the same age? Or did they

(incorrectly) get different ages?

- Sections 2.5-2.6 are well done and thoughtful.

-L406-108: The LRT test for the ATAC-seq data has the same problem as the RNA-seq analysis, which I detailed above. Further, there is no statistical test showing enrichment for promoters and distal intergenic regions (putative enhancers) in the differential peaks compared to the null expectation. Most ATAC-seq peaks fall in these regions anyway, so I think that this would be expected.

minor comments:

L89: what is "densely dissected"? Do they mean "carefully"? I would just delete these two words.

L109-116: Thank you for checking and adding these QC metrics. This text breaks up the flow of the main text and are not essential to the main findings, just necessary for someone who wants to dive deeper. Thus, I would suggest putting this in the supplementary text.

L121 (and elsewhere): there is some awkward wording. I would change this to "clustered by region", rather than "formed clusters by regions".

L122-124: this should go in the supplemental text.

L131-132: what does "highly correlated to specific brain regions" mean? How does one correlate a module with a brain region? Does this mean "enriched"? If so, statistics would be good to show here.

L133-135: This sentence should start the next paragraph, which unpacks the enrichment results.

L186: change to "but not functional categories were significantly enriched in these genes"

L188: change "cortex" to "cortical"

L195-196: Bonferroni corrected $p < 10 \times 10^{-76}$?! That's ridiculously low... how is there even power to detect this?

L286: delete "were"

L317-320: this is a strange sentence. Can the authors clarify?

The detailed point-by-point responses are provided below. The reviewers'
comments are listed below in *italicized* font and specific concern been
numbered. Our response is in red text and changes / additions to the
manuscript are given in blue text.

**Reviewer #1** (Remarks to the Author):

*I have no major additional comments on this manuscript but several*
*suggestions.*

*The writing and presentation has improved substantially but the story still*
*lacks structure. Sections summing up the relationship between expression*
*differences and known brain specific genes are interlaced with sections about*
*interindividual variability (line 190 and 332) and sections about new*
*transcripts, sex and age.*

*It would IMO be better to begin sections about finding known genes related to*
*brain structures (more concisely because its known) as validation of the study*
*samples. Then introduce the new genes showing that new information can*
*arise from the data and then end with interindividual variability (controlling for*
*sex and age) ending with a discussion on what this data may mean. A*
*question that could arise is if the new genes found here have not been found*
*before because there's higher interindividual variability for them. There's also*
*no reason to discuss sex and aging separately but as for controls to the*
*observed interindividual variability. Focusing on these*
*things as separate issues gives the manuscript an unstructured feel and leave*
*the authors open to questions about power.*

**Response:** We really appreciate the reviewer's advice regarding writing
structure. As the reviewers and editor suggested, we removed the section
regarding interindividual variability and considerably attenuated the age and
sex-associated differential gene analysis. This analysis has been moved to
the end of result section 2.1 (Line 129-140), and shortened as a paragraph.
As a result, the revised manuscript is more concise and structured.

*P222 I'm skeptical the authors can measure microglial cells given how many*
*of the cells within their samples are actually microglia.*

**Response:** As reviewer 3 suggested, we used the more stringent
permutation-based mixed-effect model to analyze the effect of age and sex
upon gene expression. These genes were no longer among the top age-
differential genes and therefore removed in the revised text.

*The ATAC seq data line p390 is still not well integrated with the RNA-Seq*
*data (do the two correlate?) is interindividual variation in ATAC-Seq correlated*
*with inter individual variation in RNA-Seq. This data now appears as a*

*sideshow (oh we also have this data) to the rest of the manuscript. It could be*
*discussed as part of the transcriptional inter-individual variability part.*

**Response:** We thank the reviewer for the suggestion. We have strengthened
the ATAC-seq dataset with samples from the hippocampal CA1 region. We
also refined the ATAC-seq data analysis (Line 312-375), and identified open
chromatin regions specific to CA1 and PCG cortex. We compared the
differential peaks with previously annotated enhancer list. As reviewers
suggested, we removed the analysis regarding inter-individual variability. In
the revised manuscript, we refined the correlation analysis, and revealed the
correlation between region-specific open chromatin (putative enhancers) and
region-related genes.

*There's still several spelling and sentence structure mishaps, that force me to*
*infer what the authors mean. The authors could let the manuscript be*
*proofread by a professional organization such as lifescienceeditors or*
*something alike? Naturally that part is only a suggestion.*

**Response:** We have carefully proofread the revised manuscript.

*Reviewer #2 (Remarks to the Author):*

*The authors have addressed most of my comments. My only remaining*
*questions are about Section 2.3, Discovering genes with high individual-*
*variation:*

*1) It is unclear how analysis is done. I understand that the authors fit two*
*models, and compare them by LRT. However, in the full model: expression ~*
*individual ID + region, how is "individual ID" coded? If it is binary coded (one*
*variable per sample), then the full model has 8 more variables (hence 8 more*
*coefficients) than the reduced model. It is a bit strange to analyze data in this*
*way, as the two models differ a lot in complexity.*

**Response:** The "individual ID" was coded as one variable with 8 levels in the
previous revision. As the reviewers suggested, we have removed this section
to make the manuscript more concise and structured.

*2) In the comparison of two models, I would think age and sex could be major*
*confounders, as they could have large impact on gene expression. So it's*
*possible that the results are largely driven by the extent by which gene*
*expression responds to age/sex, instead of "intrinsic" variability of gene*
*expression. A simple thing to do is perhaps to adjust for age/sex in the*
*regression model.*

**Response:** As the reviewers suggested, we have removed this section to
make the manuscript more concise and structured. Also, we replaced LRT
with a more stringent permutation-based mixed-effect model to analyze the
effect of age and sex. This analysis has been shortened and moved to the
end of result section 2.1 (Line 129-140).

*Reviewer #3 (Remarks to the Author):*

*This resubmission still represents a fantastic resource that the authors have*
*put together. As far as I know, it is the most comprehensive genomic atlas of a*
*rhesus brain to date. The authors have thoughtfully revised the manuscript in*
*response to my (and others') comments. However, I still have some major*
*concerns, particularly with the statistical analysis, that the authors should*
*address.*

*major comments:*

*- I agree with reviewer 1 that this would be a great publication as a resource in*
*Nat Comm (or another journal that publishes resources), but there isn't much*
*new biological insight/interest. However, I'll leave this decision up to the*
*editor.*

**Response:** Apart from confirming previously known information about the
macaque brain, this study suggested new region-specific marker genes, and
discovered novel transcripts and genes in rhesus macaque brain that
complement the current macaque genome. Our ATAC-seq data also
complement the current epigenomic knowledge of the macaque brain. We
believe that our RNA-seq and ATAC-seq dataset as resources would be
useful for a wide audience of scientist.

*- L124-126: cellular heterogeneity is clearly something the authors would like*
*to highlight here. Could the authors test this hypothesis by doing a*
*deconvolution analysis using the single-cell data available in Zhu et al, 2018,*
*Science, to test their potential interpretation here?*

**Response:** We thank the reviewer for the suggestion. We performed
deconvolution analysis based on previously published single-cell RNA-seq
dataset of rhesus macaque brain using CIBERSORT. We found that different
brain regions varied in inferred cellular composition. However, within each
brain region, the inferred cellular composition of samples from different
individual was similar, especially in the cerebral cortex. These results suggest
that heterogeneity in cellular composition might not be the source of
interindividual variation in the cerebral cortex.

- Section 2.3. There are two reasons why the authors cannot say much about
individual differences, and why I would suggest they remove this whole
section and most interpretation:

1. The authors are still severely underpowered to test for inter-individual
differences and the LRT approach doesn't help with an $n=8$

2. There are many technical reasons for differences among individuals. For
instance, the animals were sampled on different days, perhaps libraries
prepped on different days, RNA extracted on different days, slight variation in
the PMI, etc. For this reason, I don't think it is really valid to interpret this as
biological differences among individuals without clearly acknowledging that
these might be technical or without significantly increasing the sample size.

Further, if I am interpreting the authors correctly, then they are using a
likelihood ratio test to see if this model:

~ individual + tissue

fits better for a gene than this model:

~ tissue

I think this is an inappropriate way to analyze the data for the reasons I
outlined above (and because sex and age aren't included). There are many
technical, rather than biological reasons, for this to be a better fit. The authors
also omitted age and sex from this model. In sum, I would encourage the
authors to remove the sections on individual differences. It's not necessary for
the resource.

**Response:** As suggested by the reviewer and the editor, we removed the
section on individual differences and most of the interpretations regarding
individual differences.

- Section 2.4: Thank you for including the details of the LRT in the methods
(section 4.9). Having seen these details I now see that the authors are
inappropriately assuming that the repeated samples from individuals are
independent and not correlated with age and sex (which they are!). This
makes it very difficult to trust their findings.

Further, they are assuming that all of the regions within the 16 "closely related
sections" are all the same. Am I correct in reading that the authors are
comparing these two models?

~ Age + Sex

vs

~ Age (or ~Sex)

if so, then the authors needs to change the model to compare:

~ Age + Sex + Individual + Region

vs

~ Age + Individual + Region (or ~ Sex + Individual + Region)

*Otherwise they are pseudoreplicating their data and age/sex effects (which*
*could be technical) could be misinterpreted as individual differences. It would*
*be even better if the authors could include individual ID as a random effect in*
*a mixed-effects model framework.*

**Response:** We did not assume the regions within the 16 sections were all the
same. In the previous revision, we added region as variable in both models for
sections containing multiple regions. We apologize for not making it clear in
the methods of previous revision.

We greatly appreciate this excellent question. In light of this question, we
applied a mixed effect model-based method to analyze our dataset.

Mathematically, this model, called Negative Binomial mixed-effect generative
model (NBMEGM), can be written as:

$$Y_{gijkl} \sim NB(\mu_{gijkl}, \theta),$$

$$\log(\mu_{gijk}) = \alpha_{gi} + \beta_{gj} + \gamma_{gk} + \delta_{gl},$$

$$g = 1, \dots, G; i = 0,1; j = 0,1; k = 1, \dots, 16; l = 1,2; s = 1,2,3.$$

where Y_{gijkl} represent the k-th observed number of sequencing reads of
gene g from individual l at sex level i , age level j , region k , μ_{gijkl} represent
the average parameter of the Negative Binomial distribution with θ
measuring the rate of over-dispersion, $\alpha_{gi}, \beta_{gj}, \gamma_{gk}$ are fixed effects and
respectively denote the effect of age, sex and region at level i, j, k for gene g,
δ_{gl} is a random effect for gene g from individual l and follows normal
distribution $N(0, \sigma_{gl}^2)$. We here assume that Y_{gijkl} are mutually independent
from each other.

In order to avoid the cross-impact of age and sex in detecting the covariant
related genes, we implement permutation-based large-scale hypothesis test¹
to identify genes that are related to sex and age. For sex-related gene
identification, we pooled the four measurements of each gene within the j-th
age level, permuted them and assigned each two reordered measurements to
the cells (i, j) for $i = 0,1$. Here cell (i, j) means a data set with sex at level i, age
at level j. This procedure is repeated for $j = 0,1$ to complete one permutation.
We conducted all 18 permutations for each gene dataset, fit NBMEGM to
each permuted dataset by using gam method (R package mgcv) and obtain a
total of $18 \times G$ p-values. These p-values are then used to construct the
empirical null distribution. The empirical null distribution that constituted by
directly pooling the resulted p-values can always be biased, as mentioned by
these studies¹⁻³, this is mainly because this distribution is a permutation
distribution under the null hypothesis mixed with the one under alternative
hypothesis, thus could induce a larger variation. We therefore apply locfdr
method (R package locfdr) proposed by Efron et al.⁴, to adjust the bias

bringing by the nonnull p-values and to give an estimation to the real empirical
null distribution. The significant genes are then selected according to the
standard FDR method based on this estimated empirical null distribution,
using fdrci (R package, $FDR < 0.1$, $\log_2FC > 1$). Identification of age-related
genes follows the same technical line.

Results of this analysis has been shortened considerably and moved to the
end of result section 2.1 (Line 129-140).

- L709-710: did you permute ages within individuals or across them? In other
words, for each permutation, did different regions from the same individual get
the same age? Or did they (incorrectly) get different ages?

**Response:** For each permutation, different regions from the same individual
get the same age.

- Sections 2.5-2.6 are well done and thoughtful.

-L406-108: The LRT test for the ATAC-seq data has the same problem as the
RNA-seq analysis, which I detailed above. Further, there is no statistical test
showing enrichment for promoters and distal intergenic regions (putative
enhancers) in the differential peaks compared to the null expectation. Most
ATAC-seq peaks fall in these regions anyway, so I think that this would be
expected.

**Response:** As suggested, we removed the analysis regarding interindividual
variability in ATAC-seq data. We strengthened the ATAC-seq dataset with
samples from the hippocampal CA1 region. To avoid pseudo-replication of
data, we chose PCG as representative region of the cortex. We then
compared ATAC-seq peaks from CA1 and PCG, and identified open
chromatin regions specific to CA1 and PCG cortex. We annotated the function
of genes correlated with differential peaks, and compared the differential
peaks with previously annotated enhancer list using Chi-square test for
statistical significance.

*minor comments:*

L89: what is “densely dissected”? Do they mean “carefully”? I would just
delete these two words.

**Response:** As suggested, we removed these words in the revised
manuscript.

L109-116: Thank you for checking and adding these QC metrics. This text
breaks up the flow of the main text and are not essential to the main findings,
just necessary for someone who wants to dive deeper. Thus, I would suggest
putting this in the supplementary text.

**Response:** We thank the reviewer for the suggestion. This information has
been moved to the supplementary text.

*L121 (and elsewhere): there is some awkward wording. I would change this to*
*“clustered by region”, rather than “formed clusters by regions”.*

*L122-124: this should go in the supplemental text.*

**Response:** We changed the words as suggested (Line 112). Also, the
corresponding text has been moved to the supplement.

*L131-132: what does “highly correlated to specific brain regions” mean? How*
*does one correlate a module with a brain region? Does this mean “enriched”?*
*If so, statistics would be good to show here.*

**Response:** Details of this analysis were presented in methods section 4.6

**Gene module discovery:**

The gene modules were characterized based on the expression of the module
eigengene (ME), and region traits were expressed as binary matrix. Pearson
correlation between each region and each ME was calculated, and the
student asymptotic *p*.value was calculated correspondingly. The correlation
between region and ME was considered significant if the correlation
coefficient was no less than 0.8 and *p*.value was < 0.01.(Line 647-652)

In the revised manuscript, we added the corresponding correlation coefficient
and *p*.value in the text and Figure 2a for clarity.

*L133-135: This sentence should start the next paragraph, which unpacks the*
*enrichment results.*

**Response:** We moved this sentence to the beginning of the next paragraph
as suggested (Line 176-178).

*L186: change to “but not functional categories were significantly enriched in*
*these genes”*

**Response:** We revised the text as suggested (Line 225-226).

*L188: change “cortex” to “cortical”*

**Response:** This section has been removed in the revised manuscript, as
suggested. We have changed other occurrence of “cortex region” to “cortical
region” in the text.

L195-196: Bonferroni corrected $p < 10 e^{-76}$?! That’s ridiculously low... how is
there even power to detect this?

**Response:** Each individual had over 30 cortical regions, which might be the
reason behind this finding. Nevertheless, this section regarding individual
variability has been removed in the revised manuscript, as suggested.

L286: delete “were”

**Response:** We revised the text as suggested. (Line 247)

L317-320: this is a strange sentence. Can the authors clarify?

**Response:** In this paragraph, we presented several examples of genes that
were only expressed in subcortical structures like the striatum and pons.
Since the cerebral cortex is the closest brain structure to the skull, it is easily
accessible and most previous studies sampled the cerebral cortex.
Consequently, these cortex-focused RNA-seq studies could not find the
expression of subcortical-specific genes. In addition, annotation of the rhesus
macaque genome was based on these studies, and subcortical-specific genes
would also be ignored. We have revised the text for clarity.

The absence of these subcortical transcripts in the current annotation was
likely due to the fact that previous RNA-seq studies were focused on the
cerebral cortex of the rhesus macaque brain.

(Line 280-282)

**References**

- 1. Gao, X. Construction of null statistics in permutation-based multiple
testing for multi-factorial microarray experiments. *Bioinformatics* **22**,
1486–1494 (2006).
- 2. Xie, Y., Pan, W. & Khodursky, A. B. A note on using permutation-based
false discovery rate estimates to compare different analysis methods for
microarray data. *Bioinformatics* **21**, 4280–4288 (2005).
- 3. Efron, B., Tibshirani, R., Storey, J. D. & Tusher, V. Empirical Bayes
Analysis of a Microarray Experiment. *J. Am. Stat. Assoc.* **96**, 1151–1160
(2001).
- 4. Efron, B. Large-scale simultaneous hypothesis testing: the choice of a
null hypothesis. *J. Am. Stat. Assoc.* **99**, 96–104 (2004).

Reviewers' Comments:

Reviewer #1:

Remarks to the Author:

The manuscript has improved considerably. I have no further questions.

Reviewer #2:

Remarks to the Author:

The authors have addressed all my comments.

Reviewer #3:

Remarks to the Author:

The authors have done a nice job responding to my previous reviews, particularly by removing some questionable analyses from the previous version. I think this is a great resource for the NHP neurogenomics community and look forward to citing this paper and using the data in future work!

REVIEWERS' COMMENTS:

Reviewer #1 (Remarks to the Author):

The manuscript has improved considerably. I have no further questions.

Reviewer #2 (Remarks to the Author):

The authors have addressed all my comments.

Reviewer #3 (Remarks to the Author):

The authors have done a nice job responding to my previous reviews, particularly by removing some questionable analyses from the previous version. I think this is a great resource for the NHP neurogenomics community and look forward to citing this paper and using the data in future work!

Response: We are glad that the referees found the manuscript improved considerably and that they have no further questions. We thank the editors and referees for their time and effort spent in reviewing of this study.